# Direct and indirect pathways linking the Lon protease to motility behaviors in the pathogen *Pseudomonas aeruginosa*

Aswathy Kallazhi[1], Anamika Rahman[1], Ute Römling[2], Kristina Jonas[1]*

**1** Department of Molecular Biosciences, The Wenner-Gren Institute, Science for Life Laboratory, Stockholm University, Stockholm, Sweden, **2** Department of Microbiology, Tumor and Cell Biology, Karolinska Institute, Stockholm, Sweden

* kristina.jonas@su.se

## Abstract

The ATP-dependent cytoplasmic protease Lon has critical functions in protein quality control and cellular regulation in organisms across the three domains of life. In the opportunistic pathogen *Pseudomonas aeruginosa*, *lon* loss-of-function mutants exhibit multiple phenotypic defects in motility, virulence, antibiotic tolerance and biofilm formation. However, only a couple of native substrate proteins of Lon are described in *P. aeruginosa* until now and most of the phenotypes associated with Lon remain unexplained. Here, we searched for novel Lon substrates in *P. aeruginosa* by analyzing proteome-wide changes in protein levels and stabilities following *lon* overexpression. Our search yielded a large number of putative Lon substrates with diverse cellular functions, including metabolic enzymes, stress proteins and a significant fraction of motility-related proteins. *In vitro* degradation assays confirmed the metabolic protein SpeH, the heat shock protein IbpA as well as seven proteins involved in flagella- and type IV pilus-mediated motility as novel substrates of Lon. The new motility-associated substrates include both key regulators of motility (FliA, RpoN, AmrZ) as well as structural flagellar components (FliG, FliS and FlgE). Further, by isolating suppressor mutations bypassing the motility defect of *lon-* cells, we reveal that Lon-dependent degradation of the specific substrate SulA, a cell division inhibitor, is crucial for ensuring proper cell division and motility under optimal conditions. In sum, our work highlights Lon's regulatory role in degrading functional proteins involved in critical cellular processes and contributes to a better molecular understanding of the pathways underlying *Pseudomonas* pathogenicity.

## Author summary

Proteases specifically degrade other proteins inside cells and constitute promising drug targets due to their central roles in cell physiology. In the human

**Data availability statement:** All data generated or analysed during this study are included in the manuscript and supporting files. The mass spectrometry proteomics data have been deposited to the ProteomeXchange Consortium via the PRIDE partner repository with the dataset identifier PXD063095.

**Funding:** The study was funded by project grants from the Swedish Research Council (2016-03300, 2020-03545 and 2024-04942 to KJ), a future leaders grant from the Swedish Foundation for Strategic Research (SSF, FFL15-0005 to KJ), as well as funding from the Strategic Research Area (SFO) program distributed through Stockholm University (to KJ), Karolinska Institute (to UR) and the Lillian Sagens and Curt Ericssons Research Foundation (to UR). The funders had no role in the study design, data collection and analysis, decision to publish, or preparation of the manuscript.

**Competing interests:** The authors have declared that no competing interests exist.

pathogen *Pseudomonas aeruginosa*, the protease Lon has been implicated in the regulation of various processes that are linked to the ability of *P. aeruginosa* to tolerate antibiotic exposure and to establish infections. Here, we have determined the group of proteins that are directly degraded by Lon in *P. aeruginosa*. Among these proteins, we found several proteins known to be involved in swimming, swarming and twitching motility, which are critical behaviors of *P. aeruginosa* during host colonization. Further, we found that by degrading a cell division inhibitor protein, Lon ensures proper cell division, growth and motility under optimal growth conditions. Our work sheds light onto the molecular mechanisms linking the Lon protease to bacterial pathogenicity, which is critical for exploring the potential of Lon and other cellular proteases as antibiotic drug target.

## Introduction

All cellular proteins are subject to strict regulation and quality control via the proteostasis network, which involves a set of protein machines that precisely control the synthesis, maintenance and degradation of proteins. One of the major components of this network are proteases, which are tasked with the degradation of proteins. This is vital, not only to remove misfolded or aggregated proteins, but also to precisely adjust the cellular concentrations of specific proteins according to the physiological and metabolic context of the cell [1]. One major group of proteases are the AAA+ (ATPases Associated with diverse cellular Activities) proteases that unfold and degrade substrate proteins in an ATP-dependent manner [2]. In order to precisely regulate the composition of the proteome, each of these proteases degrades a distinct panel of substrates, which are recognized through short amino acid recognition sequences, called degrons, and in some cases aided by adaptor proteins [3]. Most of these proteases exhibit a high degree of specificity and there are specific sets of proteins whose stability is exclusively controlled by each protease [4–6]. However, recent studies suggest that there is some overlap between the substrate pools of different proteases, especially under stress conditions [7–9].

Lon is one of the major cytoplasmic AAA+ proteases and is highly conserved across the three domains of life. It is comprised of a homo-hexameric ring, in which each monomer consists of an N-terminal domain for substrate recognition, an AAA+ domain for substrate unfolding using ATP hydrolysis and a C-terminal protease domain containing a Ser-Lys catalytic dyad required for proteolytic cleavage [10]. Lon was first described as a protein quality control protease that degrades un- or misfolded proteins as well as incomplete proteins [11,12]. A large body of subsequent work showed that it also has critical regulatory functions, thereby impacting a variety of important cellular processes [13]. In the model organisms *Escherichia coli*, *Caulobacter crescentus* and *Bacillus subtilis*, previously identified Lon substrates include regulators of stress responses, cell differentiation, and cell cycle progression, metabolic and structural proteins as well as antitoxin proteins of toxin-antitoxin systems [14–20]. For many years, relatively few native Lon substrates were known; however,

recent developments in mass spectrometry-based proteomics approaches aided the search for protease substrates and resulted in the identification of hundreds of putative Lon substrates in selected model organisms [4,15,20]. In spite of these advances, the nature of Lon substrates still remains poorly studied in many other species, including bacterial pathogens.

In the opportunistic pathogen *Pseudomonas aeruginosa*, proteases including Lon have been shown to affect a variety of pathways that are linked to its pathogenicity [21–26]. *P. aeruginosa* is a common cause of community and hospital-acquired acute and chronic infections of the urinary tract, the airways and other bodily sites [27]. Due to its low nutritional needs, its high adaptability and the ability to develop antibiotic resistance, *P. aeruginosa* remains difficult to control. It is an ESKAPE pathogen listed by the WHO in the Bacterial Priority Pathogens List (WHO BPPL) [28,29]. Besides other virulence factors, a crucial contributor to the pathogenicity of *P. aeruginosa* is its ability to form biofilms [30]. Although biofilm formation is a predominantly sessile lifestyle, different forms of motility are required to successively initiate biofilm formation and shape the biofilm structure. It uses three major forms of motility called swimming, swarming and twitching, where the first two rely on rotations of its polar flagellum, while the latter relies on extensions and retractions of a type IV pilus. All three forms of motility have been shown to be critical for host colonization and cell invasion [31].

Based on phenotypic analyses of a *lon* loss-of-function mutant in *P. aeruginosa*, Lon has been reported to affect the three major types of motility as well as biofilm formation, virulence in amoeba and mouse lung infection models and antibiotic susceptibility [21–23,26]. Despite these wide-reaching effects, only a couple of proteins have been verified as direct Lon substrates in *P. aeruginosa* to date, namely the SOS-induced cell division inhibitor SulA that is also a well-studied substrate of *E. coli* Lon [18,32] and the RNA-binding protein Hfq [32]. Additionally, Lon has also been shown to be involved in the LasI/LasR quorum sensing pathway, where LasI degradation was shown to be dependent on Lon *in vivo* [33]. Given the multiple phenotypes exhibited by *lon* mutants and the wide range of Lon substrates that have been described in other model bacteria, Lon is expected to contribute to the regulation of a substantial number of proteins in *P. aeruginosa*.

To uncover novel substrates of Lon in *P. aeruginosa* and to unravel the molecular basis of previously reported phenotypes of *lon* loss-of-function, we used mass spectrometry-based quantitative proteomics to search for proteins with altered steady-state levels and stabilities in response to *lon* overexpression. Among a large group of putative substrates, we verified a metabolic protein SpeH, the heat shock protein IbpA and seven proteins associated with flagella- or type IV pili-mediated motility as direct Lon substrates. Furthermore, we demonstrate that Lon is critical for proper cell division and swimming motility in *P. aeruginosa* under optimal conditions by maintaining low levels of the cell division inhibitor SulA.

## Results

### Quantitative proteomics experiment reveals putative substrates of Lon

Previous work has shown that Lon substrates can be identified based on changes in their steady-state levels and protein stabilities in strains that either lack or overexpress *lon* [15,20]. Hence, we sought to find novel Lon substrates in *P. aeruginosa* by identification of proteins that exhibit reduced abundance and stability in *lon*-overexpressing cells (Fig 1A). We generated a *lon* overexpression strain (*lon*OE) of *P. aeruginosa* clone C clinical urine isolate 8277 containing plasmid-borne *lon* under the control of an arabinose-inducible promoter. Immunoblot analysis confirmed an approximately 2-fold upregulation of Lon levels within 90 min of induction (S1A Fig) in comparison to a vector control (VC) strain carrying the empty pJN105 vector.

To assess relative protein stabilities between the *lon*OE and VC strain, we grew these strains in the presence of L-arabinose for 90 min before protein synthesis was shut off by addition of spectinomycin, to then follow the decay of proteins over time by collecting samples at 0, 30 and 60 minutes. We chose a spectinomycin concentration of 1 mg/mL,

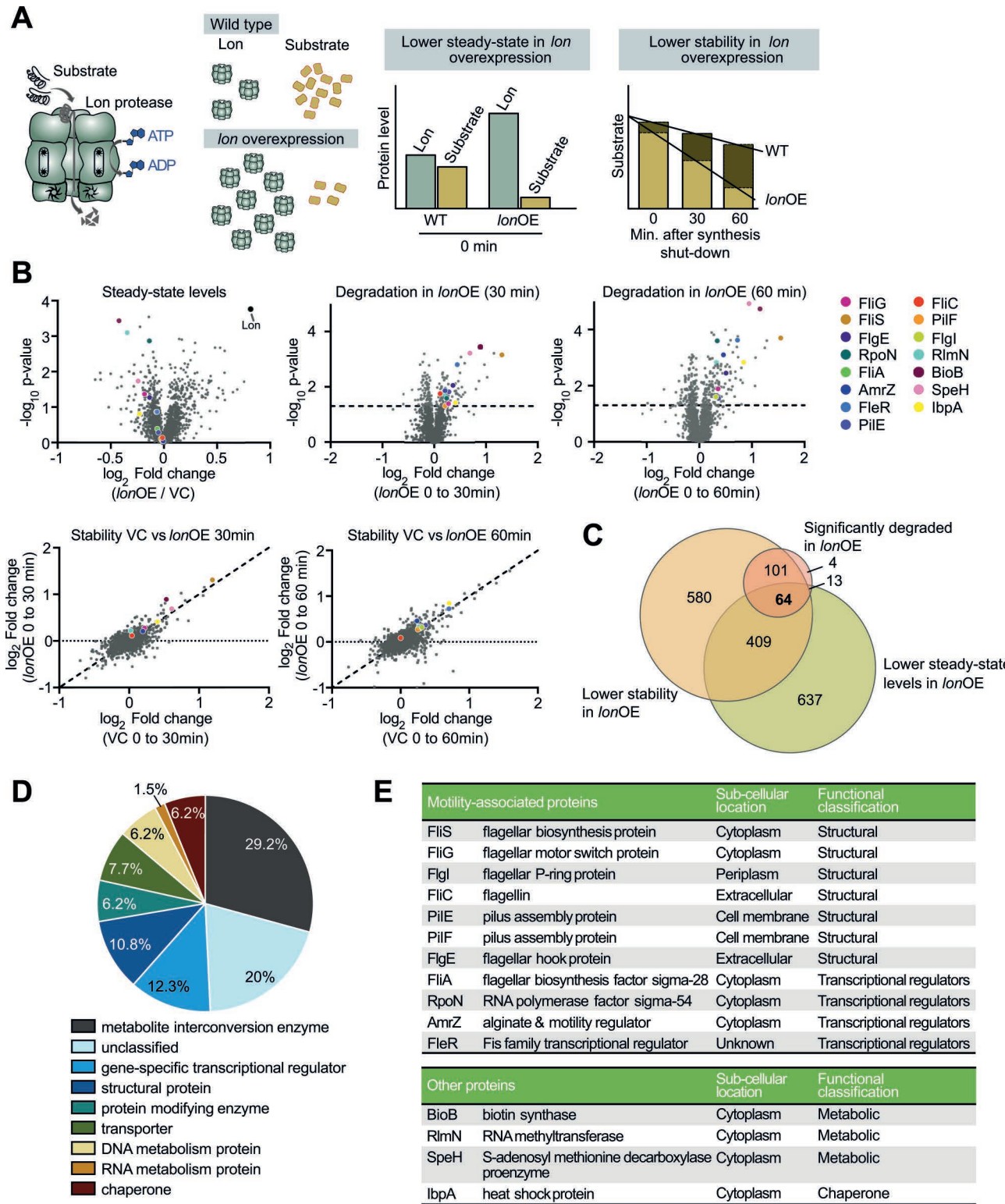

**Fig 1. Global quantitative proteomics experiment reveals putative substrates of Lon.** (A) Strategy of searching for Lon substrate candidates based on changes in protein levels and stabilities upon *lon* overexpression (WT – wild type, *lon*OE – *lon* overexpression). (B) Volcano and dot plots representing the proteomics data in *P. aeruginosa* clone C isolate 8277. Volcano plots show proteome-wide effects of *lon*OE in comparison to the

vector control (VC) after 90 min of *lon* induction (*top left*) as well as proteome-wide changes in protein levels 30 min (*top center*) and 60 min (*top right*) after spectinomycin-induced synthesis shut-down in the *lon*OE strain in comparison to the 0 min time point. The dotted line indicates $-\log_{10}$ p-value of 1.3 (corresponding to a p-value of 0.05). The dot plots below show comparisons of proteome-wide protein stabilities between the *lon*OE strain and the VC. $\log_2$ fold-changes between 0 min and 30 min (*bottom left*) as well as 0 min and 60 min (*bottom right*), respectively, for the VC and the *lon*OE strains were plotted against each other. Motility-associated proteins and other selected top-ranked proteins as well as Lon protease itself are marked. p-values were calculated using a two-tailed t-test. Three independent biological replicates of 0 and 60 min and two independent biological replicates of 30 min for each of the control and *lon* overexpression strains were used. (C) Venn diagram showing selection of substrate candidates by applying three different criteria. Criterion I, green circle — proteins showing a lower steady-state level in *lon*OE compared to the vector control. Criterion II, orange circle — proteins showing a larger reduction between 0 min and 30 or 60 min in *lon*OE compared to the VC. Criterion III, red circle — proteins that are significantly degraded in *lon*OE, *i.e.*, showing a significant (p < 0.05) downregulation between 30 or 60 min and 0 min. (D) Pie chart of the functional classification analysis of the 64 proteins satisfying all the criteria in Fig 1C. (E) Table of motility-associated proteins and other selected proteins found in the final list of Lon substrate candidates in Fig 1D along with their primary function in the cell, subcellular location and their functional classification. FliA and FlgE did not satisfy all three criteria, but were included in this list due to their functions in motility and their previous connections with Lon in other bacteria [15,34].

at which culture growth was strongly reduced, indicating efficient inhibition of protein synthesis (S1B Fig). Time point 0 min was sampled before spectinomycin addition and thus provided information about the steady-state levels of proteins after 90 min of induction of *lon* expression with L-arabinose. SDS-PAGE analysis showed no striking changes in the total protein fractions across the collected samples (S1C Fig). However, immunoblot analysis with the Lon specific antiserum revealed a mild reduction in Lon levels in both strains over time, indicating that Lon itself is subject to degradation (S1C Fig), a result that is consistent with a previous report [35].

To quantify proteome-wide protein levels and stabilities, we used a quantitative mass spectrometry approach coupled to TMT (Tandem Mass Tag) isobaric labelling and HiRIEF (High Resolution Iso-Electric Focusing) fractionation. A total of 2264 proteins were detected in all samples. Consistent with the immunoblot data (S1A Fig), Lon levels were significantly upregulated in the *lon*OE strain compared to the VC control strain (Fig 1B).

By comparing the 0 min time points of the *lon*OE and VC strains, we found that 1123 proteins were downregulated in response to *lon* overexpression (Fig 1B). To analyze Lon-dependent changes in protein stabilities, we first analyzed the $\log_2$ fold-changes between 0 min and 60 min as well as 0 min and 30 min for both the VC and the *lon*OE strains and compared the $\log_2$ fold-changes between both strains. A larger $\log_2$ fold change in the *lon*OE strain at 30 or 60 min compared to the VC indicated lower protein stability in response to *lon* overexpression (Fig 1B, bottom), and we found 1154 proteins to show this behavior. Finally, we applied a more stringent criterium of selecting only proteins that showed significant degradation in *lon*OE, *i.e.,* a significant downregulation between 0 min and 30 or 60 min after spectinomycin addition (Fig 1B), which resulted in 182 proteins. It is noteworthy that a fraction of proteins showed an increase in abundance after inhibition of protein synthesis by spectinomycin (Fig 1B). This might be the result of some ongoing protein synthesis even after addition of spectinomycin (S1B Fig).

In order to shortlist candidates for potential Lon substrates, we combined all three criteria and selected proteins that showed reduced steady-state levels in the *lon*OE strain compared to the VC strain as well as significantly unstable proteins in *lon*OE (*i.e.*, showing a significant decrease between 30 or 60 min and 0 min after spectinomycin addition) that also show reduced stability compared to the VC strain. This resulted in a final list of 64 proteins which satisfied all of these criteria and were considered to be potential candidates for further investigation (Fig 1C). Among the previously reported substrates, SulA and LasI were not detected, while Hfq showed mildly reduced levels upon *lon* overexpression, but did not satisfy the other criteria we applied.

Analysis of the annotated functions of these proteins showed that they represented all major functional categories, including a large fraction of metabolic proteins, transcriptional regulators and structural proteins, but also many uncharacterized proteins (Fig 1D). Metabolic proteins such as BioB, RlmN and SpeH as well as the heat shock protein IbpA, which is a known Lon substrate in *E. coli* [36], were among the top candidates in the putative substrates list sorted by highest steady-state level changes and were chosen for further analysis (Fig 1E). Interestingly, nine of the 64 shortlisted putative

substrates were all associated with flagella- or type IV pilus-mediated motility, six of which were classified as structural proteins and three as transcriptional regulators. Additionally, we found the flagellar transcriptional regulator FliA and the flagellar hook protein FlgE, both of which are Lon substrates in other bacteria [15,34], in the list of proteins downregulated by *lon*OE. The high representation of motility-related proteins among the shortlisted candidates was promising, considering that motility is critical for *P. aeruginosa* virulence and has been shown to be affected by the Lon protease [26]. Hence, we also selected this group of motility-associated proteins for further investigation (Fig 1E).

### *In vitro* degradation assays confirm Lon proteolysis of nine novel substrates

To test whether the selected substrate candidates are directly degraded by Lon, *in vitro* degradation assays using purified components were performed. Out of the eleven putative Lon substrates that were annotated to have functions in motility, five are cytoplasmic, including the alternative sigma factors RpoN and FliA, the pleiotropic transcriptional regulator AmrZ and two structural proteins of the flagellum (FliG, FliS), while the flagellar hook protein FlgE is extracellular. BioB, RlmN, SpeH and IbpA are all cytoplasmic proteins. We selected these proteins for their ease of purification, when compared to membrane-bound proteins. Some strains of *P. aeruginosa*, including clone C 8277, harbor a second gene, hereafter called FliS2 (SG17M genome locus tag: K0E51_RS23370), adjacent and homologous to canonical FliS (SG17M genome locus tag: K0E51_RS23365). Since the mass spectrometry analysis might have failed to discriminate peptides from these two FliS proteins, FliS2 was also included to be tested due to its homology to FliS. Further, we included the previously validated *Pseudomonas* Lon substrate SulA in our analysis [32].

First, in order to validate the activity of our purified Lon protease from *P. aeruginosa*, we monitored degradation of the model substrate β-casein [12]. Degradation of β-casein was observed in an ATP-dependent manner indicating that the purified Lon is enzymatically active (S2A Fig). As expected, we also observed fast *in vitro* degradation of SulA in the presence of ATP and creatine kinase that regenerates ATP, while remaining stable in the absence of ATP and creatine kinase (Fig 2A), confirming that this protein is efficiently degraded by *Pseudomonas* Lon [32]. Despite being strong candidates according to the proteomics data, BioB and RlmN did not show degradation by Lon *in vitro* (S2B and S2C Fig). This could mean that these proteins still undergo Lon-dependent degradation *in vivo*, but require additional factors which are absent in our *in vitro* system. SpeH (also known as SpeD2), a 17 kDa S-adenosyl methionine decarboxylase proenzyme involved in an alternative spermidine biosynthesis pathway showed two bands on SDS-PAGE, as it is known to undergo autocatalysis into two chains which then form a hetero-tetramer [37,38]. Both of these bands showed robust degradation by Lon *in vitro* (Fig 2A), while IbpA exhibited a subtle but reproducible level of degradation (Fig 2A), which matches the comparably slow Lon-mediated degradation previously observed for *E. coli* IbpA [36]. Next, we tested the seven motility proteins in our *in vitro* degradation system. Notably, all of them showed robust degradation by Lon in the presence of ATP (Fig 2B), but no notable degradation in the absence of ATP. Together, these data demonstrate that all seven motility-associated proteins as well as IbpA and SpeH are *bona fide* substrates of Lon in *P. aeruginosa*, and also indicate that our proteomics-based search was successful in identifying novel Lon substrates.

### *lon* disruption does not affect the steady-state levels of motility-associated substrates, but causes strong accumulation of SulA

According to our proteomics data, the newly validated substrate proteins are affected by Lon *in vivo* under *lon*-overexpressing conditions (Fig 1). To assess how much Lon contributes to regulating its substrate proteins in wild-type cells grown under optimal conditions, we wanted to compare the levels and stabilities of these proteins in a *lon* loss-of-function mutant to the wild type background. Since a *lon* mutant of *P. aeruginosa* clone C strain 8277 was not available, and since we did not succeed in generating a new *lon* deletion mutant in this background, we turned for the remaining experiments of this study to *P. aeruginosa* PAO1. This strain possesses Lon and most validated substrate proteins identical to clone C strain 8277, with the exceptions of FliS, which shares 76% similarity between the strains and FliS2, which

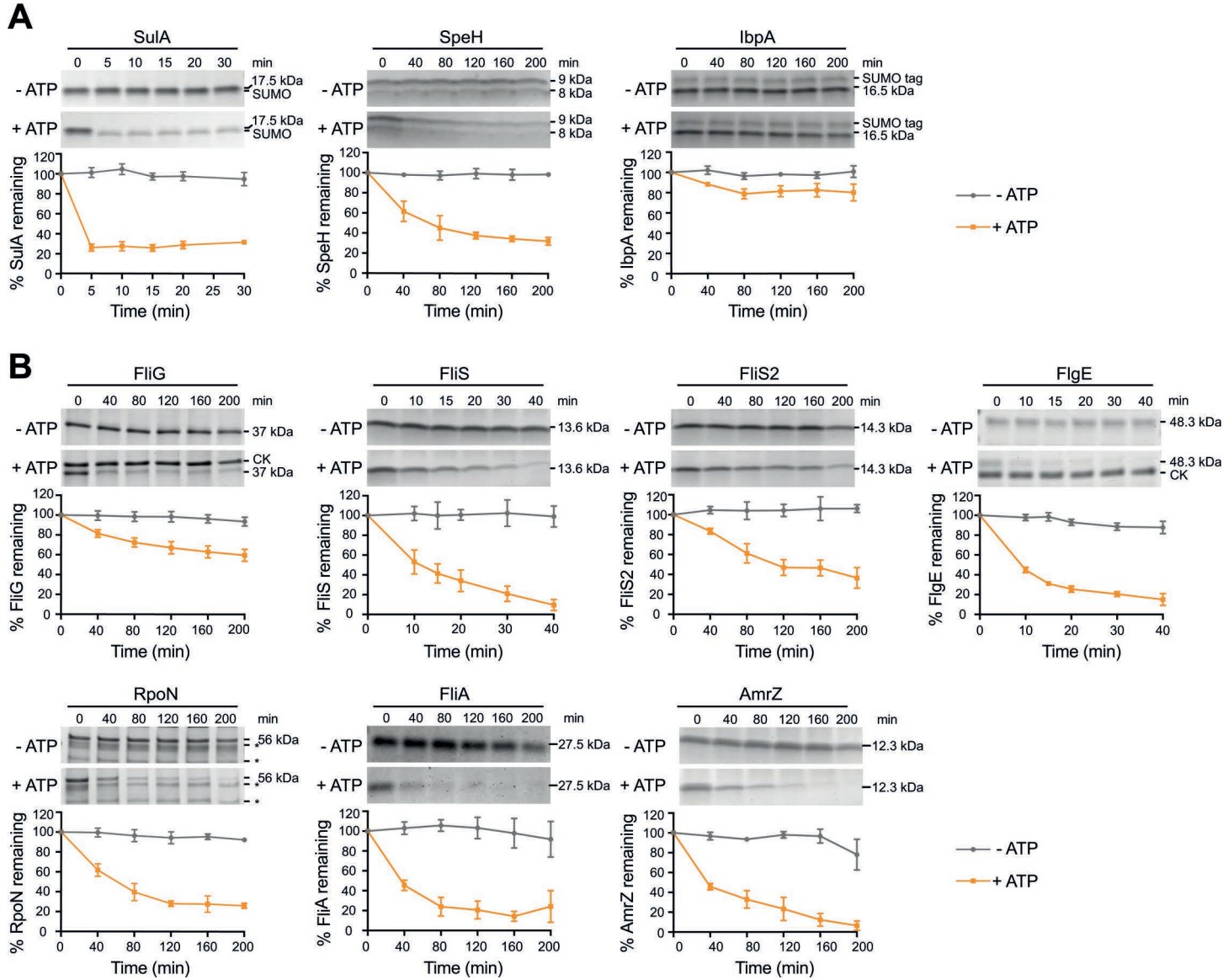

**Fig 2.** *In vitro* **degradation assays confirm seven motility-associated proteins as well as SpeH and IbpA as novel Lon substrates.** (A) *In vitro* degradation assays of the previously reported Lon substrate SulA as well as the new substrates SpeH and IbpA in the presence (orange) and absence (grey) of ATP and creatine kinase (CK). 0.4 µM of $Lon_6$ hexamer and 4 µM of substrate were used in all assays. Quantification of each of the degradation assays were done using Image Lab 6.1 and plotted using the arithmetic mean ± SD of the quantified values of remaining substrate normalized relative to corresponding Lon levels for each time point. Each graph represents quantifications of at least three independent experiments. One representative SDS-PAGE gel of the assay is shown for each protein. (B) *In vitro* degradation assays with quantifications as in (A), but with the motility-associated substrates FliG, FliS, FliS2, FlgE, RpoN, FliA and AmrZ. 0.4 µM of $Lon_6$ hexamer and 3 µM of substrates were used, with the exception of FliG that was added at a concentration of 1.4 µM. Asterisks (*) depict unspecific bands.

is not present in PAO1. We again used quantitative MS-based proteomics, but this time to specifically analyze the steady-state levels and stabilities of the new Lon substrates as well as SulA between the PAO1 wild type strain and a *lon-* strain, in which the *lon* gene has been disrupted by a transposon insertion [39]. SulA was either not detected or present at very low levels in the wild-type replicates, but was present at high levels in all replicates of the *lon-* strain, indicating that Lon is

critical for limiting SulA levels in wild-type cells during optimal conditions through degradation (Fig 3A). In contrast to SulA, none of the other validated substrates, showed a clear change in steady-state levels in the *lon-* strain (Fig 3A).

To also assess the stability of the newly identified substrates in the wild-type and *lon-* strains, we monitored the decay of these proteins after spectinomycin-induced protein synthesis shut-off. This experiment showed that FliG and FlgE were significantly stabilized in the *lon-* strain compared to the WT, indicating that Lon contributes to their degradation under optimal conditions (Fig 3B). In the cases of FliS, RpoN, SpeH and IbpA, we observed notable protein degradation in the wild type within 60 min after spectinomycin addition, which was however not significantly affected by absence of Lon in the *lon-* strain. Finally, FliA and AmrZ showed little to no degradation even in the wild type strain within 60 min after protein synthesis shut-off, indicating that these proteins might be protected from proteolysis under the chosen condition (Fig 3B).

Collectively, these results show that while Lon is critical for maintaining low cellular levels of SulA, absence of Lon proteolysis does not cause major changes in the abundance of the other substrate proteins. This result might be due to compensatory effects, either by other proteases or through transcriptional regulation, as well as mechanisms protecting substrates from degradation under the chosen conditions, as previously reported for FliA in *E. coli* that is stabilized by its anti-sigma factor FlgM during optimal growth [34].

## *lon* disruption leads to defects in growth, morphology and swimming motility

With the increased knowledge about Lon substrates in *P. aeruginosa*, we wondered if we can elucidate the molecular basis of the previously reported phenotypes of the *lon-* mutant. First, we wanted to re-assess some of the reported phenotypes of the *lon-* mutant in comparison to the wild type, *lon* overexpression as well as a complemented *lon-* strain containing a plasmid-borne arabinose-inducible copy of *lon*. Immunoblot analysis confirmed that Lon was absent in the *lon-* strain, but present in the WT and the complemented *lon-* strain (S3 Fig). Consistent with previous work [26], we observed that the *lon-* strain showed reduced swimming motility in soft agar (Fig 4A) as well as reduced twitching motility (Fig 4B) and impaired biofilm formation when quantifying bacterial attachment in microtiter plates using crystal violet staining (Fig 4C). Additionally, we observed a lesser degree of pigmentation of the *lon-* strain, indicative of reduced secretion of the secondary metabolite pyocyanin (Fig 4D). Importantly however, while the swimming and pigmentation defects could be complemented by expressing plasmid-borne *lon*, this was not the case for the biofilm and twitching phenotypes. In these cases, complementation of the *lon* disruption mutant strain with plasmid-borne *lon* exhibited even lower levels of twitching and biofilm compared with the vector control. *lon* overexpression (WT P$_{ara}$ *lon*) had no significant effect on any of the phenotypes (Fig 4A–D).

We reasoned that changes in growth rate and cell morphology might contribute to the observed Lon-dependent swimming defect. Hence, we also recorded growth curves of the different strains and analyzed cell morphology by phase contrast microscopy. These data show that the *lon-* strain exhibited a mild growth defect (Fig 4E) and also showed clear cell division defects resulting in cell elongation (Fig 4F), consistent with previous data [21]. Both these phenotypes were fully complemented by providing *lon in trans*.

Together, our phenotypic analyses confirm the previous finding that Lon is required for proper growth, cell division and swimming motility. In contrast, however, the previously reported phenotypes in twitching and biofilm formation could not be clearly attributed to Lon.

## Suppressor mutations in *sulA* restore motility of the *lon* disruption strain

Based on the phenotypic results, we reasoned that the swimming defect of the *lon-* mutant is either caused by one or several of the motility proteins that are Lon substrates, or stabilization of another substrate that affects motility more indirectly, for example through defects in cell division and slower growth.

PLOS Pathogens

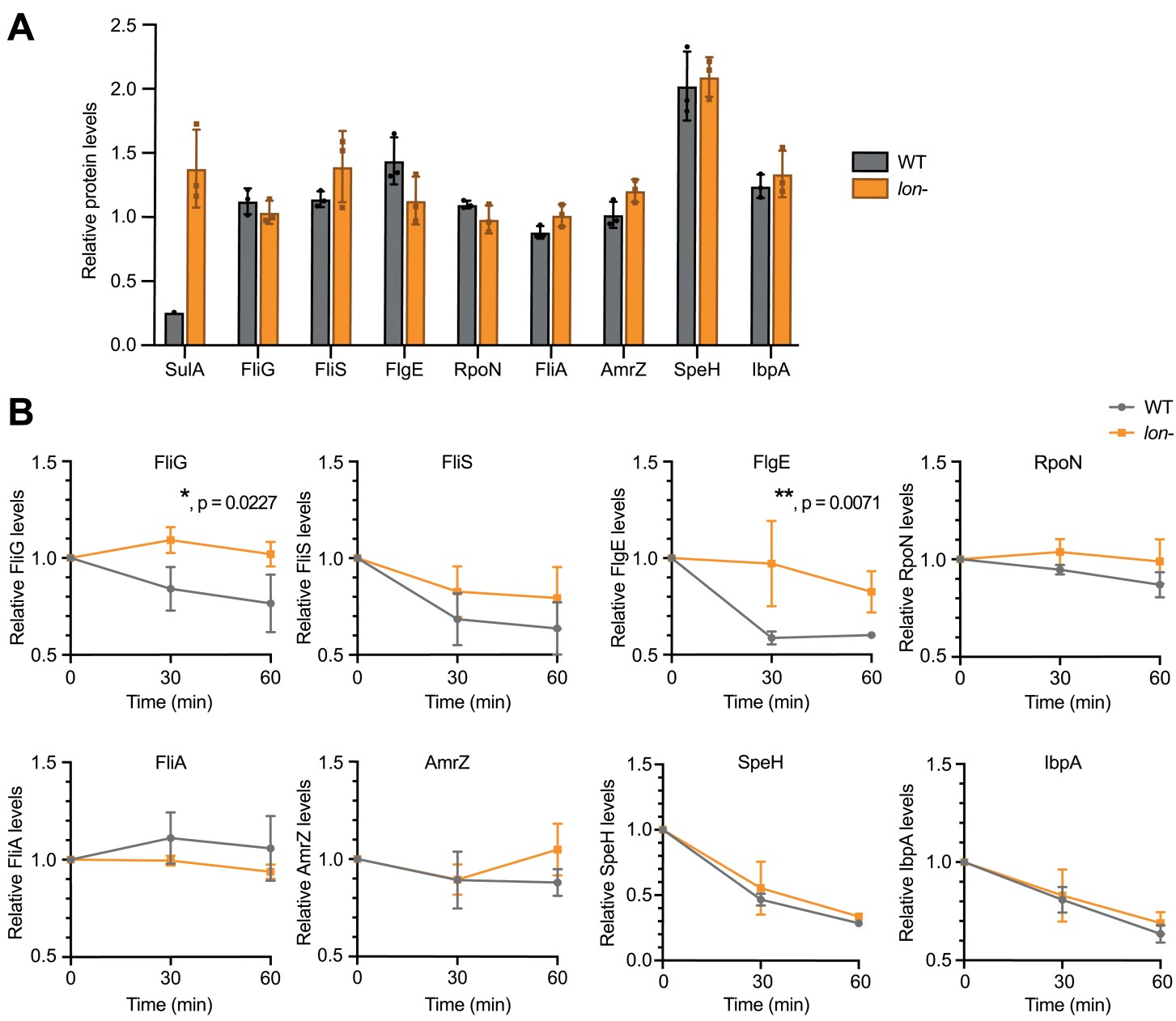

**Fig 3. Steady-state levels and stability of Lon substrates in a *lon* disruption strain.** (A) Relative steady-state levels of various Lon substrates in PAO1 WT (grey) and *lon-* (orange) backgrounds as determined by quantitative proteomics. Relative means ± SD of three independent replicates shown, with the exception of SulA in WT that was not detected in two of the replicates. (B) Protein stabilities of the newly identified Lon substrates in WT and *lon-* as determined by quantitative proteomics. Changes in protein levels over 0, 30 and 60 min after spectinomycin-induced protein synthesis shut-off are shown for the WT (grey) and the *lon-* (orange) mutant. Plotted points represent the mean values ± SD for three independent replicates relative to time point 0. A simple linear regression analysis shows that the slopes of the fitted lines significantly differ for FliG (-0.003919 for WT and 0.0003186 for *lon-*, p = 0.0227, *) and FlgE (-0.01380 for WT and -0.002926 for *lon-*, p = 0.0071, **). For all other proteins, the differences were not significant (p > 0.05).

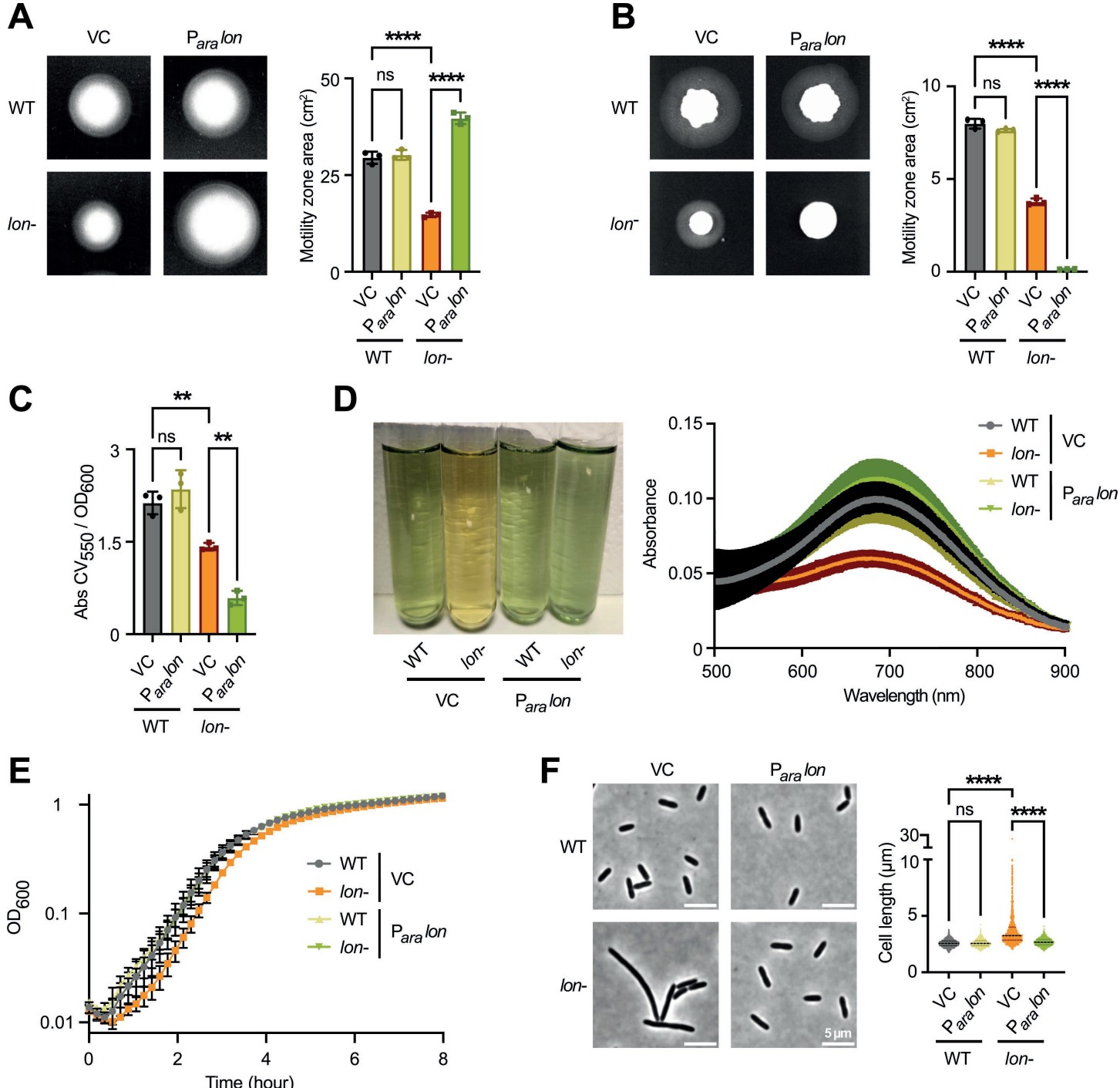

**Fig 4. *lon* disruption leads to defects in growth, morphology and swimming motility.** (A) Swimming motility in soft agar with arabinose after 40–48 hours at 25°C for the strains shown. The means of motility area (cm²) ± SD of three independent biological replicates plotted for each strain. Statistical significance was calculated using one-way ANOVA (Tukey's multiple comparisons test) for comparison of pairs: WT VC vs WT P*ara lon,* p = 0.8999 (not significant, ns); WT VC vs *lon-* VC and *lon-* VC vs *lon-* P*ara lon,* p < 0.0001, ****. (B) Twitching motility in agar plates with arabinose after 8 days at 25°C of the strains shown. Means of motility zone area ± SD of three independent biological replicates plotted. Statistical significance was calculated using one-way ANOVA (Tukey's multiple comparisons test) for comparison of pairs: WT VC vs WT P*ara lon,* p = 0.0948 (not significant, ns); WT VC vs *lon-* VC and *lon-* VC vs *lon-* P*ara lon,* p < 0.0001, ****. (C) Absorbance of crystal violet stained biofilms grown for 8 hours and measured at 550 nm normalized against cell growth at OD$_{600}$. The means ± SD of three independent biological replicates plotted for each strain. Statistical significance was calculated using

one-way ANOVA (Tukey's multiple comparisons test) for comparison of pairs: WT VC vs WT $P_{ara}$ lon, p = 0.5083 (not significant, ns); WT VC vs lon- VC, p < 0.0079, **; lon- VC vs lon- $P_{ara}$ lon, p < 0.0030, **. (D) Pyocyanin produced by late exponential cultures of the four strains grown at 37°C as visualized by supernatants. Right — Means ± SD of absorbance scans of the supernatants from three biological replicates at 500–900 nm. (E) Growth curves of the four strains grown at 37°C and measured at $OD_{600}$. Means ± SD of three independent biological replicates plotted. (F) Phase-contrast microscopy images of the four strains when grown at 37°C to logarithmic phase. Right — Violin plot shows quantifications of 2850 cells for each strain from three biological replicates. Statistical significance was calculated using one-way ANOVA (Tukey's multiple comparisons test) for pairs: WT VC vs WT $P_{ara}$ lon, p = 0.9963 (not significant, ns); WT VC vs lon- VC and lon- VC vs lon- $P_{ara}$ lon, p < 0.0001, ****.

To uncover the genetic cause of the motility defect of the lon- strain, we decided to take an unbiased approach, in which we isolated suppressor mutations restoring motility in the lon- mutant. For this, we inoculated soft agar motility plates with the lon- strain and briefly exposed the plates to UV light to promote mutagenesis. After 48–72 hours of incubation, we observed suppressor mutants forming sectors of increased swimming diameters that emerged from the swimming zone of the original lon- strain (Fig 5A). Independent suppressor mutants from different biological replicates were isolated and swimming motility of the suppressors was confirmed to be significantly higher than that of the lon- strain (Fig 5A). Whole genome sequencing of six of these suppressors (referred to as Sup1–6 hereafter) alongside the WT and lon- strains revealed that each of them contained a mutation in the sulA gene when compared to the original lon- strain (Fig 5B). Sup1, 2, 3 and 4 carried the same point mutation leading to a H127L amino acid substitution in SulA, Sup5 contained another point mutation leading to a D109V amino acid substitution and Sup6 harbored a deletion in sulA (Δ133–148 nt) resulting in a frameshift mutation, which in turn caused a truncation of the corresponding SulA protein. All suppressor mutants restored not only motility, but also normal cell division (Fig 5C), suggesting that all of them are loss-of-function mutations that disrupt SulA's function as a cell division inhibitor. On the other hand, they did not restore pyocyanin production and rescued growth only partially, indicating that Lon might be affecting these phenotypes through other substrates (S4A and S4B Fig).

To analyze more carefully how the suppressor mutations affect SulA function, we ectopically expressed SulA$^{H127L}$ (Sup1–4), SulA$^{D109V}$ (Sup5) and the truncated SulA$^{fs}$ (frameshift in Sup6) with a 3xFLAG-tag in the wild-type strain background and compared their effects on cell division to a strain expressing 3xFLAG–tagged wild-type SulA (F-SulA). While the wild-type F-SulA caused very strong cell filamentation, consistent with SulA's function as cell division inhibitor [40], expression of the mutants led either to no or moderate filamentation (Figs 5D and S5A), showing that they either completely or partially lost their ability to block cell division. Interestingly, the moderate cell filamentation caused by F-SulA$^{D109V}$ was not seen in the corresponding suppressor mutant in the lon- background, which completely restored motility and cell division (Fig 5C). This can likely be explained by higher protein levels when expressed from the pBAD plasmid compared to the native expression level from $P_{sulA}$. Probing the levels of the different SulA variants with an anti-FLAG antibody showed that F-SulA$^{H127L}$ and F-SulA$^{D109V}$ were clearly detectable, but at reduced levels compared to wild-type SulA (Fig 5E). The truncated SulA$^{fs}$ variant was not detected, likely due to the truncated protein being unstable.

Next, we wondered how expression of the different SulA variants in the wild-type background would affect motility. While expression of F-SulA resulted in almost complete inhibition of motility, expression of F-SulA$^{H127L}$ affected motility only mildly compared to the vector control strain (Figs 5F and S5B). Expression of the F-SulA$^{D109V}$ variant, which caused moderate cell filamentation (Fig 5D), also impaired motility, but not as much as expression of wild-type F-SulA (Figs 5F and S5B). In the absence of the inducer arabinose, leaky expression of F-SulA was still enough to strongly inhibit motility, while no significant swimming defects were observed under this condition for the strains harboring F-SulA$^{H127L}$ or F-SulA$^{D109V}$ (S5C Fig). Together, these results demonstrate that SulA-induced cell filamentation correlates with reduced soft-agar motility, and suggest that the SulA suppressor mutants restore motility of the lon- mutant by reducing cell filamentation.

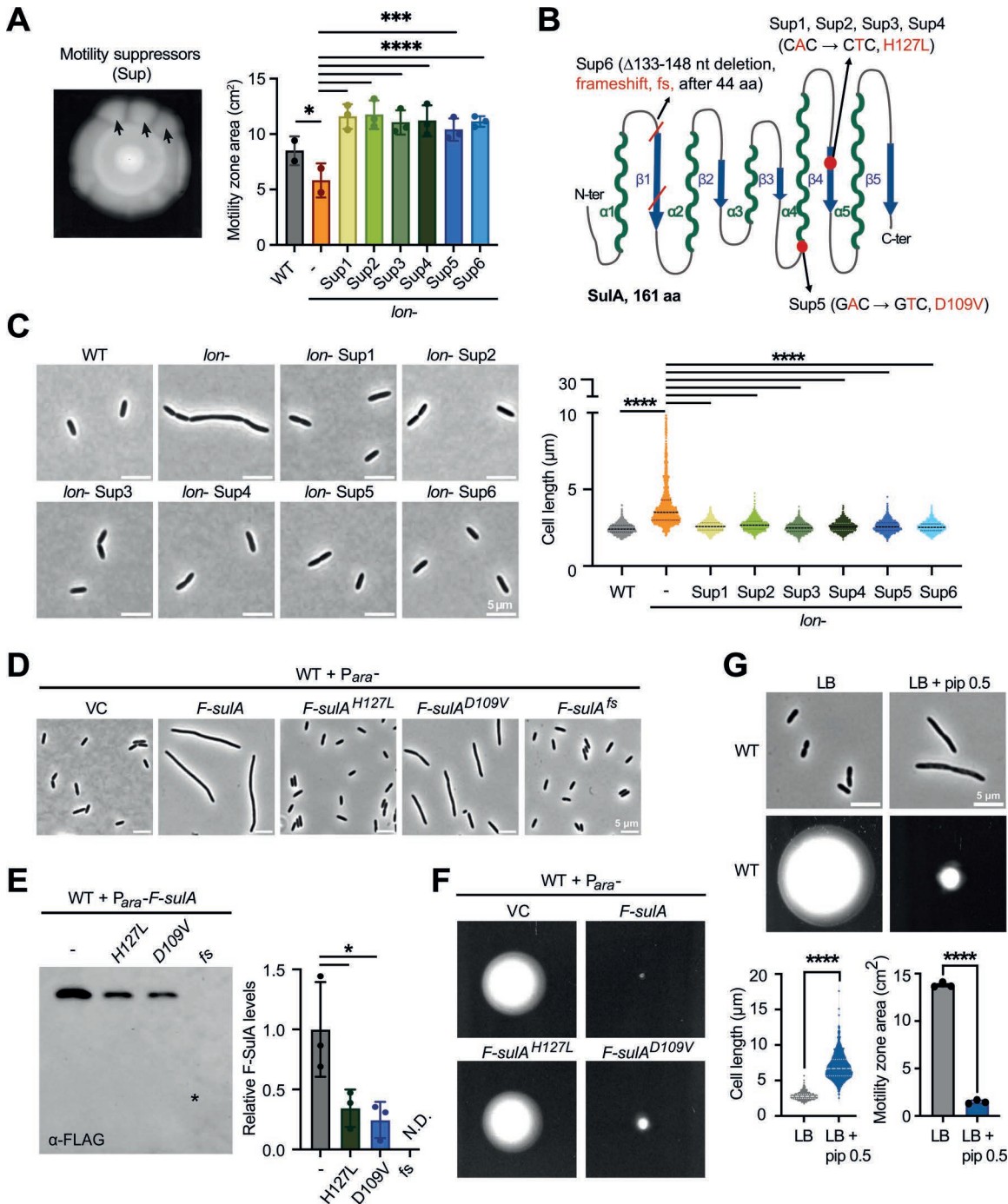

**Fig 5. Suppressor mutations in *sulA* restore motility of the *lon* disruption strain.** (A) Image showing suppressor zones emerging from the motility zone of the *lon*- strain on a soft agar plate incubated at 25°C for 48–72 hours after UV-irradiation. *Right* — The means of quantified motility zone area (cm²) ± SD of two (WT, *lon*-) or three independent biological replicates (Sup1–6) plotted. Statistical significance was determined using one-way ANOVA (Tukey's multiple comparisons test) for pairs: WT vs *lon*-, p = 0.0230, *; *lon*- vs Sup1, 2, 3, 4, 6, p < 0.0001, ****; *lon*- vs Sup5, p = 0.0003, ***. (B) Illustration of the location and nature of the six suppressor mutations (*red*) on a simplified structure of SulA depicting alpha helices (green) and beta sheets (blue). (C) Phase-contrast microscopy images of WT, *lon*- and the six suppressor strains after arabinose induction grown at 37°C to logarithmic phase. *Right* — Violin plot shows the quantification of 2850 cells for each strain from three independent biological replicates. Statistical significance was calculated using one-way ANOVA (Tukey's multiple comparisons test) for pairs: WT vs *lon*- and *lon*- vs Sup1, 2, 3, 4, 5, 6, p < 0.0001, ****. (D) Phase-contrast microscopy images of the strains shown after arabinose induction, grown at 37°C to logarithmic phase. F – 3xFLAG tag. fs – frameshift. (E)

Representative immunoblot of the strains shown after arabinose induction, using an anti-FLAG antibody. '*' on the blot indicates the expected location of F-SulA[fs]. *Right* — Quantification of immunoblots based on three independent biological replicates. Statistical significance was calculated using one-way ANOVA (Tukey's multiple comparisons test) for pairs: F-SulA vs F-SulA[H127L], p = 0.0297, *; F-SulA vs F-SulA[D109V], p = 0.0144, *. N.D – not detected. (F) Swimming motility in soft agar with arabinose after 40–48 hours at 25°C for the strains shown. (G) *Top* — Phase-contrast microscopy images of WT and WT treated with 0.5 µg/mL piperacillin for two hours grown at 37°C to logarithmic phase. *Middle* — Swimming motility in soft agar without and with 0.5 µg/mL piperacillin after 40–48 hours at 25°C. *Bottom left* — Violin plot showing the quantification of 2250 cells for each condition from three independent biological replicates. Statistical significance was calculated using unpaired t-test for LB vs LB + pip 0.5, $p < 0.0001$, ****. *Bottom right* — The means of motility zone area (cm$^2$) ± SD of three independent biological replicates plotted. Statistical significance was calculated using an unpaired t-test LB vs LB + pip 0.5; $p < 0.0001$, ****. Pip – piperacillin.

In order to further establish the connection between cell division inhibition and soft agar motility, we opted to inhibit cell division in a SulA-independent way and tested how this affects motility. Piperacillin is a penicillin-like antibiotic known to inhibit cell division by binding FtsI, preventing peptidoglycan synthesis and inhibiting FtsZ ring formation [41,42]. Consistent with previous data [43], we found that it induces strong cell filamentation of *P. aeruginosa* at a sub-inhibitory concentration of 0.5 µg/mL (Fig 5G), at which growth is not notably affected (S6 Fig). Importantly, adding this piperacillin concentration to swimming agar plates resulted in a severe motility defect (Fig 5G), confirming an inverse correlation between cell filamentation and soft-agar motility. Based on this result, we suggest that Lon indirectly promotes motility by lowering SulA levels and hence ensuring proper cell division. Hence, accumulation of SulA in the *lon-* results in cell division defects that cause the formation of filamentous cells that are less motile in soft agar.

## A secondary mutation in *pilI* causes twitching and biofilm defects in the *lon-* strain

Having sequenced the genome of the *lon-* strain, we also searched for mutations that may explain the twitching and biofilm phenotypes of this strain that were not possible to complement with plasmid-borne *lon* expression (Fig 4B and 4C). Indeed, we found a single base pair deletion causing a frameshift mutation in the *pilI* coding region that was absent in the parental PAO1 strain. This frameshift mutation results in a stop codon immediately downstream of the single base pair deletion and thus in the translation of a short, truncated PilI peptide (Fig 6A). *pilI* encodes a putative type IV fimbrial biogenesis protein and is part of the chemotaxis-like gene cluster that mediates twitching motility [44] and is also predicted to affect biofilm formation [45]. To test whether the background mutation in *pilI* is the reason for the defects in these phenotypes observed in the *lon-* strain, we provided a functional copy of the *pilI* gene on a plasmid to the *lon-* strain. Indeed, the defect in biofilm formation (Fig 6B) was fully recovered with a functional *pilI* gene despite the strain still lacking functional Lon. We also assayed twitching motility and observed that expression of wild-type *pilI* significantly increased the twitching of the *lon-* strain, but not to wild type levels (Fig 6C). This could be due to the SulA-dependent cell division defect of the *lon-* strain partially contributing to the lowered twitching motility. These results indicate that the biofilm and twitching defects of the *lon-* strain can be attributed to a large extent to the background mutation in *pilI*. It also shows that the PilI protein of the pilus biogenesis gene cluster, which remains to date poorly characterized, plays an important role in biofilm formation and type IV pilus-mediated twitching of *P. aeruginosa*.

## Discussion

The Lon protease is increasingly recognized as a critical regulatory protease that contributes to the coordination of cellular behaviors and stress responses by specifically degrading functional proteins [13,14,20,46,47]. In *P. aeruginosa*, Lon has been associated with numerous cellular processes that are connected to its pathogenic and highly adaptable life style [21–23,26,33]. However, the mechanistic basis by which Lon affects these processes remained poorly understood, partly due to limited knowledge about native substrate proteins in this organism. In this study, we filled these knowledge gaps by describing novel substrates of *Pseudomonas* Lon with diverse functions and by linking the Lon-dependent degradation of specific substrate proteins to previously reported phenotypes (Fig 7).

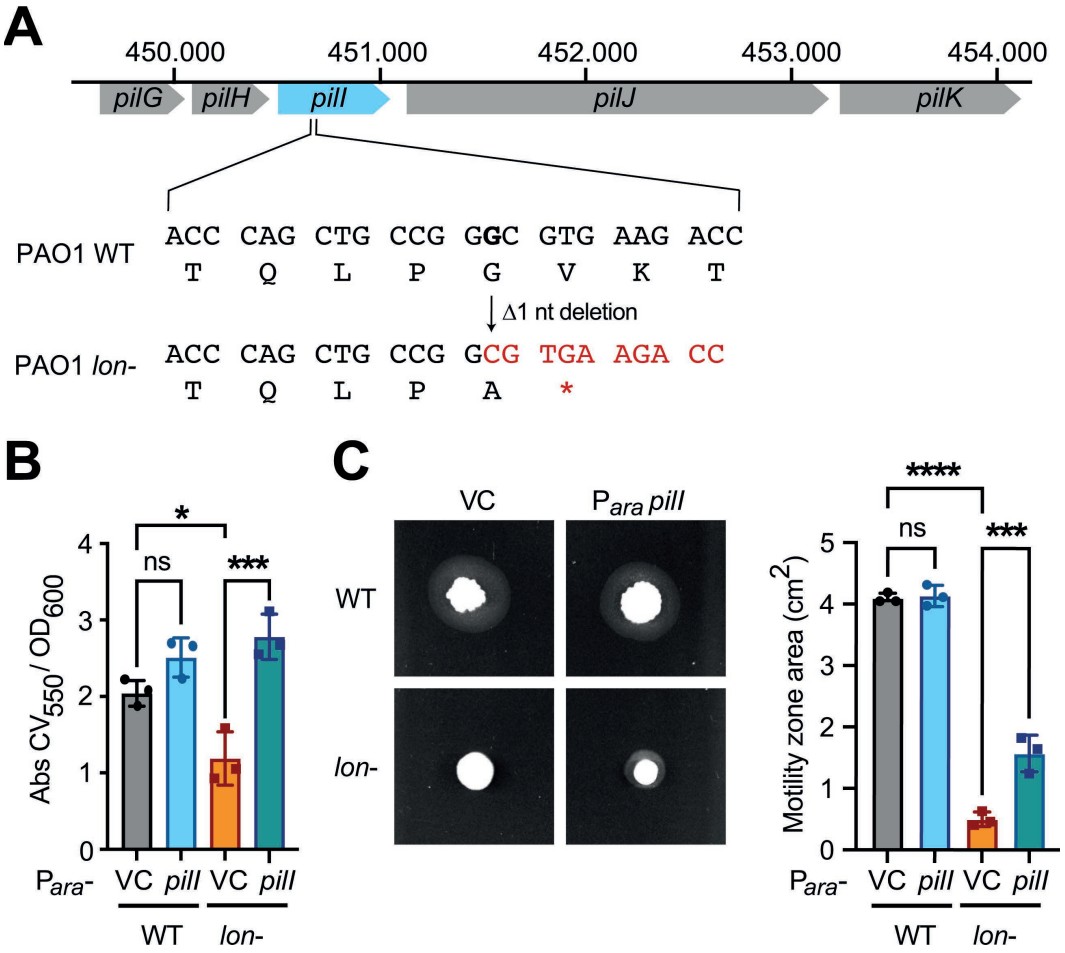

**Fig 6. A secondary mutation in the *pilI* gene causes defects in twitching and biofilm formation in the *lon-* strain.** (A) Illustration of the single base pair deletion mutation on the *pilI* gene on the *pilGHI* gene cluster on the genome causing a frameshift leading to a stop codon immediately downstream. (B) Absorbance of crystal violet stained biofilms grown with arabinose for 8 hours and measured at 550 nm normalized against cell growth at $OD_{600}$. The means ± SD of at least three independent biological replicates plotted. Statistical significance was calculated using one-way ANOVA (Tukey's multiple comparisons test) for comparison of pairs: WT VC vs WT $P_{ara}$ *pilI*, p = 0.2359 (not significant, ns); WT VC vs *lon-* VC, p < 0.0222, *; *lon-* VC vs *lon-* $P_{ara}$ *pilI*, p < 0.0005, ***. (C) Twitching motility in agar plates with arabinose after 5 days at 25°C of the strains WT VC, WT $P_{ara}$ *pilI*, *lon-* VC and *lon-* $P_{ara}$ *pilI*. Means of motility area (cm²) ± SD of at least three independent biological replicates plotted. Statistical significance was calculated using one-way ANOVA (Tukey's multiple comparisons test) for comparison of pairs: WT VC vs WT $P_{ara}$ *pilI* p = 0.9939 (not significant, ns); WT VC vs *lon-* VC p < 0.0001, ****; *lon-* VC vs *lon-* $P_{ara}$ *pilI* p = 0.0005, ***.

Using a proteomics-based approach, we identified a large group of candidate substrate proteins based on changes in their steady-state levels and stabilities in cells overexpressing *lon*. We confirmed nine of them as *bona fide* Lon substrates. Interestingly, despite showing robust Lon-mediated degradation *in vitro*, the levels of the validated Lon substrates, with the exception of SulA, were not notably affected by *lon* loss-of-function. In case of the motility proteins, their Lon-dependent degradation was also dispensable for proper motility under optimal growth conditions. This suggests that absence of Lon is either compensated, for example by transcriptional changes or degradation by other proteases, or that Lon proteolysis of these substrates is more critical under alternative growth conditions. Our experiments were conducted in exponentially growing cultures in rich media conditions. Under this condition, motility is known to be favored [48] and thus, the need for degrading motility proteins is expected to be low. However, at the entry to stationary phase and under

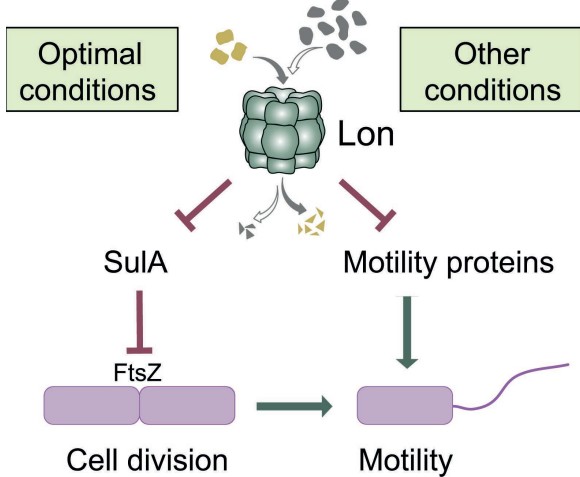

**Fig 7. Lon-dependent control of flagella-based motility in _Pseudomonas aeruginosa._** Under optimal conditions Lon is critical for maintaining low SulA levels which allows cell division to take place, which in turn is needed for proper motility. Lon also degrades various motility proteins, including FliG, FliS, FliS2, FlgE, FliA, RpoN and AmrZ. The degradation of these proteins likely contributes to the downregulation of motility under specific growth conditions, for example during the transition from a motile into a sessile life style.

various conditions during host colonization and invasion, motility is downregulated and subpopulations of cells will instead transition to sessile lifestyles [49]. It is possible that Lon proteolysis of motility proteins is critical during such growth transitions. Alternatively, Lon proteolysis might play a homeostasis-promoting role and remove flagella proteins that are in stoichiometric excess and not bound to their proper binding partners, as previously suggested [50]. Similarly, IbpA degradation is expected to be particularly needed for homeostatic purposes or under conditions during stress recovery, when there is a need for downregulating IbpA. Deciphering the exact conditions under which Lon-mediated degradation of the newly identified substrates takes place is however challenging due to the difficulty in monitoring protein degradation precisely during growth transitions and in individual cells.

In spite of significant differences in the flagellar regulon cascade and amino acid sequences of homologous proteins, Lon was shown to degrade motility-associated proteins in several other bacteria. The flagellar hook protein FlgE and the flagellar sigma factor FliA, which we have identified here as _Pseudomonas_ Lon substrates, have also been described as Lon substrates in _E. coli_ [15,34]. In _C. crescentus,_ the flagellar hook length regulator FliK, the flagellar assembly regulator FliX as well as FlgE were reported as Lon substrates [20] and in _B. subtilis_ Lon degrades the master flagellar activator SwrA [19]. These data suggest an evolutionarily conserved role of Lon in degrading motility-associated proteins independent of protein identity and precise amino acid sequences (as exemplified by FliS and FliS2).

In addition to revealing novel direct substrates of _Pseudomonas_ Lon, our study explains some of the previously reported phenotypes of _lon_ loss-of-function. Our data demonstrate that stabilization and strong accumulation of the cell division inhibitor SulA in _lon-_ cells results in cell division defects and cell filamentation, which in turn impairs motility (Fig 7). The filamentous morphology of cell-division inhibited cells is expected to hamper their penetration through the soft agar. Moreover, given that _P. aeruginosa_ is monotrichous, its single flagellum is expected to operate less efficiently on cells with strongly increased cell mass.

Although SulA was previously shown to be a substrate of _Pseudomonas_ Lon [32], it is noteworthy that it possesses a C-terminus that is clearly dissimilar from the C-terminal degron sequence of _E. coli_ SulA that mediates recognition by Lon [51]. The physiological consequences of Lon-dependent degradation of SulA remained also unclear in _P. aeruginosa_ and have not been linked to motility behaviors and other phenotypes of the _lon-_ strain. Our data show that SulA stabilization

PLOS Pathogens

in this strain is the major cause of its swimming motility and cell division defects and it is possible that other phenotypes observed in *lon-* mutants, such as reduced virulence in infection models or increased antibiotic susceptibility can be explained by the presence of excess SulA [21,22]. It is notable that the suppressor mutations D109V and H127L in SulA were both in close proximity to regions (Gln106 to Cys108 and Val120 to Gly125) predicted to be involved in FtsZ binding [52]. Hence, these mutations might disrupt the interaction between SulA and FtsZ, thus resulting in a complete or partial loss of SulA's inhibitory function on cell division. The importance of Lon in limiting SulA levels in wild type cells is interesting and potentially suggests a role for Lon in adjusting cell division under certain environmental or stress conditions, for example, caused by exposure to antibiotics. In this scenario, reduced availability or activity of Lon in response to certain conditions would trigger cell filamentation. Based on prior studies it was suggested that SulA-induced cell filamentation constitutes a survival mechanism that helps bacteria to protect themselves against phagocytosis by immune cells [53] and to accelerate resistance development through homologous recombination [54].

In conclusion, our study has shed light onto the cellular roles of Lon in the opportunistic pathogen *P. aeruginosa* by revealing novel substrates through quantitative proteomics and by explaining several of the previously reported phenotypes of *lon* loss of function. Our findings reveal a role of Lon in degrading motility-associated proteins and its importance in limiting the amounts of the cell division inhibitor SulA, and we also report a background mutation in the *lon-* strain, thus clarifying previously reported phenotypes in twitching and biofilm formation [22,26]. Furthermore, our study has revealed many other putative Lon substrates that remain to be further studied in the future. Altogether, our work illustrates how cytoplasmic proteases such as Lon contribute to the regulation of functional proteins thereby affecting critical cellular activities and also contributes to a better understanding of the molecular pathways underlying *P. aeruginosa* pathogenicity.

## Materials and methods

### Bacterial strains

All bacterial strains used in this study are listed in S1 Table.

### Growth conditions

*P. aeruginosa* strains Clone C 8277 and PAO1 strains were routinely grown in LB medium at 37°C while shaking at 200 rpm unless otherwise indicated. For all experiments involving plasmid induction, 0.2% arabinose was added to both the expression strain as well as the control strain with the empty vector, for at least 90 min before sampling. When needed, the medium was supplemented with either gentamicin (30 µg/ml) or tetracycline (50 µg/ml).

The salt-inducible BL21-SI/pCodonPlus *E. coli* strain for protein expression was grown at 30°C using LBON/2xYTON no salt medium. When necessary, media was supplemented with antibiotics at following concentrations (concentration in liquid/solid media in µg/ml): gentamicin (15/15), kanamycin (30/50) and chloramphenicol (20/40). *E. coli* DH5α strain used for cloning was grown at 37°C.

### Plasmid construction

All plasmids used in this study are listed in S2 Table.

**Expression plasmids.** For constructing plasmid pJN105:*lon*, the *lon* gene from Clone C isolate 8277 was cloned into a pJN105 vector with gentamicin resistance cassette for selection, under the control of an L-arabinose-inducible promoter. The gene was inserted between restrictions sites of NheI and PstI on the plasmid backbone through restriction cloning. The same was done for constructing pJN105:*pilI*. For *sulA* and its mutants, a 3x FLAG tag was inserted in the pJN105 plasmid between the NheI and NdeI restriction sites and the *sulA* mutants were cloned into the NdeI site to have the 3xFLAG tag at the N-terminus of the proteins. All primers used are mentioned in S3 Table. The reaction was used to transform *E. coli* DH5α competent cells which were selected on LB plates containing gentamicin. The plasmid

was isolated and sequence-verified. The pJN105 empty vector (VC) and cloned overexpression plasmids were inserted through electroporation into electrocompetent *P. aeruginosa* cells of either WT or *lon-* background to create the vector control, overexpression or complementation strains. The successful transformants were selected on LB plates containing gentamicin after overnight growth at 37°C.

**Protein expression plasmids.** Plasmids for protein expression were generated by amplifying the corresponding gene from chromosomal DNA of Clone C strain 8277 using primer pairs as listed in S3 Table and subsequent cloning of the insert into a pSUMO-YHRC vector backbone by Gibson assembly. The plasmid was amplified as two fragments from the original using primer pairs oMJF34/oMJF36 and oMJF37/oMJF38 (S3 Table) disrupting the kanamycin resistance cassette. The two fragments and the gene to be cloned were then joined using Gibson assembly and the reaction was used to transform chemically competent DH5α cells which were selected on LB plates containing kanamycin. The plasmid was then extracted and sequence-verified. The primer pairs used for cloning of each gene are as listed in S3 Table.

## Sample collection and preparation for proteomics experiments

For the *lon* overexpression proteomics experiment, samples were collected and prepared as follows: Cultures of *P. aeruginosa* Clone C 8277 containing either the empty vector pJN105 or pJN105:*lon* were grown in triplicates of 200 mL LB cultures containing gentamicin for plasmid maintenance, into exponential phase cultures by back diluting 3–4 times. After collecting 1 mL pre-induction samples for immunoblot analysis, 0.2% arabinose was added to both cultures and grown at 37°C for 90 min. 1 mg/ml spectinomycin was added to both the cultures and 50 mL of the culture was immediately retrieved into pre-cooled 50 mL tubes, spun down in a pre-cooled centrifuge and the supernatant discarded. The pellet was frozen using liquid nitrogen and stored at –20°C. $OD_{600}$ values were measured throughout the experiment at hourly intervals. This was repeated at 30 min and 60 min after addition of spectinomycin. Three replicates of 0 and 60 min samples and two replicates of 30 min samples were collected.

For the *lon-* proteomics experiment, *P. aeruginosa* strain PAO1 WT and *lon* transposon mutant *lon-* were grown in triplicates. After back-diluting 3–4 times, the exponential phase cultures were treated with 1 mg/mL spectinomycin and samples were collected at 0, 30 and 60 min as described before. Three replicates of 0, 30 and 60 min samples were collected.

Immunoblot samples were taken at all time points. The samples were later transferred to –80°C for storage until delivery to the MS facility.

## Quantitative proteomics

Proteomics analyses were conducted by the Clinical Proteomics Mass Spectrometry Facility, Karolinska Institute/Karolinska University Hospital/Science for Life Laboratory. The samples were lysed and subjected to protein digestion using trypsin followed by multiplex TMT (Tandem Mass Tag) isobaric labelling and HiRIEF (High Resolution Iso-Electric Focusing) fractionation before loading onto the LC-MS/MS. The multiplexing allowed all samples in each proteomics set to be analyzed simultaneously. The TMT16-plex was used for *lon*OE proteomics (16 samples) which included 3 replicates each of time points 0 and 60 min and two replicates of time point 30 min for each strain background. TMT18-plex was used for *lon-* proteomics (18 samples) which included 3 replicates each of time points 0, 30 and 60 min for each strain background. The reference genome used for identifying the proteins from unique peptides was that of *P. aeruginosa* Clone C SG17M (NCBI RefSeq: JALF01000000) for *lon*OE proteomics and *P. aeruginosa* PAO1 (NCBI RefSeq: NC_002516.2) for *lon-* proteomics. The source data for *lon*OE proteomics is available as S1 Data and the source data for *lon-* proteomics is available as S2 Data in supporting information.

## Proteomics data analysis

In the *lon*OE proteomics experiment, a total of 2264 proteins were detected in all samples. The proteins were selected using three criteria: based on reduction in steady-state levels, reduction in stability over time in *lon*OE compared to VC

and significant instability in *lon*OE. The data was analyzed for steady-state level changes by calculating the $\log_2$ fold-change between the 0 min samples of *lon*OE and VC for each protein and all proteins which show reduced relative levels in *lon*OE were chosen. The $\log_2$ fold-change between 0 min and 60 min as well as 0 min and 30 min of each protein of each strain was analyzed and all proteins which show greater fold change in *lon*OE over time in 30 or 60 min were selected. Finally, in order to shortlist candidates with the most significant instability in *lon*OE, proteins with the highest $\log_2$ fold-change between 0 min and 30 min or 60 min with $p < 0.05$ were selected. All criteria were independently applied on the original list of 2264 total proteins detected. In order to shortlist as many relevant candidates as possible, the selection was not stringent and all the proteins that satisfied these criteria were included without setting more cut-offs. Venn diagram was created using https://www.deepvenn.com [55]. The functional analysis of shortlisted candidates was performed using the ontology 'Protein class' in the PANTHER 19.0 database on www.pantherdb.org [56] and the unclassified proteins were further assigned functions based on The Pseudomonas Genome Database available on https://www.pseudomonas.com [57].

In the *lon-* proteomics experiment, the proteins of interest as shown in Fig 3 were selected for analysis. The steady-state level changes for each protein in WT and *lon-* was compared using the protein level at 0 min in the two strains. The relative stability was analysed by dividing the 30 min and 60 min values of protein levels by the corresponding 0 min value for each protein in each strain background.

## Protein purification

Purification of proteins was adapted from Holmberg et al. [58]. BL21-SI/pCodonPlus cells were transformed using the plasmids mentioned in the S2 Table by electroporation and selected on LBON agar plates supplemented with kanamycin (Kan) and chloramphenicol (Chlor). Pre-cultures (LBON or 2xYTON + Kan + Chlor) were inoculated with about 20 colonies and cultivated at 30°C overnight. One litre of 2xYTON + Kan + Chlor was innoculated by 1:100 dilution of the pre-culture and cultured until an approximate $OD_{600}$ of 1.0. The expression was started by addition of 0.5 mM IPTG and 0.3 M NaCl (final concentrations). Incubation continued at 30°C for 4 hours and cells were harvested subsequently by centrifugation (6 800 × g, 4°C, 10 min) and cell pellets stored at −80°C.

For purification, pellets were resuspended in HNG10Im (40 mM HEPES–KOH pH 7.5, 500 mM NaCl, 10% glycerol, 10 mM imidazole), supplemented with 1 mM PMSF, 1 mg/mL Lysozyme and 3 μL Benzonase/10 mL and topped up to 30 mL total volume. Cells were then lysed by 2–3 passes through an EmulsiFlex-C3 high-pressure homogenizer and peak pressure was kept between 25 000 and 30 000 psi. Lysate was cleared by centrifugation at 32 500 × g at 4°C for 1.5 h. Tagged proteins were bound to 3 mL (=1.5 mL bed volume) pre-equilibrated Talon SuperFlow Metal Affinity Resin from TaKaRa per liter culture on ice for 30 min with shaking. After washing 5 times with approximately 50 mL HNG10Im, bound proteins were eluted using HNG + 250 mM imidazole and fractions with protein concentrations ≥0.2 mg/mL were pooled. For 6 × His-SUMO tag removal, 4 μg/mL Ulp1–6 × His was added and imidazole was removed in parallel by dialysis against HNG (40 mM HEPES-KOH pH 7.5, 500 mM NaCl, 10% glycerol). Tag depletion was achieved by binding to 1 g dry Protino Ni-IDA beads and flow through and/or wash containing purified protein was collected. Protein concentration was checked afterwards via SDS-PAGE (Bio-Rad 4–20% Mini-PROTEAN TGX Stain-Free protein gel) and InstantBlue protein stain (Expedeon) or ReadyBlue (Sigma-Aldrich) and quantified using Bio-Rad ImageLab 6.0.1. When necessary, proteins were concentrated using a Pall Advanced Centrifugal Device or an Amicon Ultra Filter with a pore size around one-third of the respective protein. Before storage, 1 mM DTT and 1 mM EDTA was added to the protein. Proteins were snap-frozen in liquid nitrogen and stored at −80°C.

**IbpA purification.** The purification protocol was the same as for others except for the buffers since IbpA is an aggregation prone protein and requires different purification buffers, as mentioned in [36]. LG-5 buffer containing 50 mM HEPES–KOH (pH 8.0), 4 M urea, 1 mM BME, 400 mM potassium glutamate, 5 mM imidazole was used for wash and 50 mM HEPES–KOH (pH 8.0), 3 M urea, 1 mM BME, 400 mM potassium glutamate, 500 mM imidazole, 10% glycerol was

used for elution. The dialysis was done against buffer with 50 mM HEPES–KOH (pH 8.0), 2 M urea, 10% glycerol, 200 mM potassium glutamate, 1 mM BME and the storage buffer was 50 mM HEPES–KOH (pH 8.0), 800 mM potassium glutamate, 20% sucrose, 1 mM BME.

### *In vitro* degradation assay

*In vitro* degradation assays were performed as described previously [20]. The reaction was carried out in Lon reaction buffer (25 mM Tris-HCl pH 8.0, 100 mM KCl, 10 mM $MgCl_2$, 1 mM DTT) using 0.4 µM $Lon_6$ hexamer, respective amounts of substrate as mentioned in the figure legends and an ATP regeneration system (4 mM ATP, 15 mM creatine phosphate, 75 µg/mL creatine kinase). The reaction mix and the ATP regeneration system were prepared separately and pre-warmed to 37°C. The reaction was started by adding the ATP regeneration system to the reaction mix. Samples were taken at indicated time points and quenched by 1 volume of 2 × Laemmli SDS loading buffer and snap frozen in liquid nitrogen. Samples were heated at 65°C for 10 min and separated by SDS-PAGE (Bio-Rad 4–20% Mini-PROTEAN TGX Stain-Free protein gel) visualized by InstantBlue protein stain (Expedeon) or ReadyBlue (Sigma-Aldrich) and quantified using Bio-Rad ImageLab 6.0.1. Substrate levels were normalized to the Lon levels of the respective time points.

### Immunoblot analysis

For whole cell extract analysis, 1 mL culture samples were collected at the respective time points and cell pellets were obtained by centrifugation. Cell pellets were resuspended in 200 µL of 1 × SDS sample buffer per $OD_{600}$ 1.0, to ensure normalization of the samples. Samples were boiled at 98°C for 10 min and run on an SDS-PAGE using Mini-PROTEAN TGX Stain-Free gels (usually 4–20%, Bio-Rad). The proteins were transferred to nitrocellulose membranes by a semi-dry blotting procedure as per manufacturer guidelines. The protein gels and membranes were imaged using a Gel Doc Imager before and after the transfer, respectively, to assess equal loading of total protein as well as the quality of the transfer.

Membranes were blocked for 1 h at room temperature in 5% skim milk powder in TBS-Tween (TBST) and protein levels were detected using the anti-Lon (1:10000 dilution; kindly provided by R.T. Sauer) or anti-FLAG M2 (1:5000, Sigma-Aldrich) primary antibodies in 3% skim milk powder in TBST. Secondary antibodies, 1:5000 dilutions of anti-rabbit or anti-mouse (Thermo Fisher Scientific) and SuperSignal Femto West (Thermo Fisher Scientific) were used to detect primary antibody binding. Immunoblots were scanned using a Chemidoc (Bio-Rad) system or a LI-COR Odyssey Fc system. Relative signal intensities were quantified using the Image Lab software package (Bio-Rad).

### Motility assays

Swimming motility assays were performed on plates with 0.3% agar added to media containing 1% tryptone and 1% NaCl. Twitching motility was assayed on LB plates containing 1% agar. When needed, 0.2% arabinose and 30 µg/ml of gentamicin was added. The plates were dried for 4h and inoculated with 1 µL of *P. aeruginosa* cultures that were diluted to $OD_{600} = 0.1$. For swimming, the culture was inoculated 3 mm into the vertical centre of the agar. For twitching, the culture was inoculated all the way to the bottom of the agar, between the plate and the agar. The swimming motility plates were incubated at room temperature (approximately 25°C) for 40 – 48 hours for various experiments. The twitching motility plates were incubated at room temperature (approximately 25°C) for five to eight days. The plates were imaged using a ChemiDoc under the setting: Blots – Colorimetric. The motility zone area was quantified using the application Fiji (ImageJ).

### Plate reader-based growth curve measurements

Overnight cultures were grown and diluted to an $OD_{600}$ of 0.05 in LB with the addition of gentamicin and arabinose when applicable. 200 µL of the dilution were added into sterile 96 well transparent plates. LB medium (with or without antibiotic

Pathogens

and/ or inducer when applicable) were used as blank in multiple wells. The plate was set up in a Spark microplate reader (Tecan) at 37°C for an $OD_{600}$ measurement every 10 min for 24 hours with shaking.

## Biofilm assays

Overnight cultures were grown and diluted 100× in LB with the addition of gentamycin and arabinose when applicable. 200 μL of the dilution were added into sterile 96 well transparent plates. LB medium (with or without antibiotic and/ or inducer when applicable) were used as blank in multiple wells. The plate was incubated for 8 hours at 37°C and the $OD_{600}$ was measured at the end of the incubation. The liquid culture was poured out and washed gently by immersing in water twice. All work from this point was carried out in a ventilated hood. After removing residual water, 200 μL of 0.1% crystal violet solution was added to each well and incubated at room temperature for 15 min. The crystal violet was discarded and the wells were washed again with water twice. The crystal violet staining the biofilms was then solubilised using 200 μL of 10% acetic acid and incubated for 15 min. The solution from each well was transferred to a fresh 96-well plate and the absorbance at 550 nm was measured using a SpectraMax i3x multi-mode microplate reader.

## Pyocyanin production

Cultures were grown overnight and diluted to a starting $OD_{600}$ of 0.05 and grown for 4–5 hours until stationary phase $OD_{600}$ of about 3.0, when WT cultures develop a green colour. The cultures were spun down and the supernatant of each strain was extracted. 100 μL of the supernatant from each replicate of each strain was transferred to a 96-well plate in three or four technical replicates and an absorbance scan from 450 to 900 nm corresponding to pyocyanin secretion, was recorded on a Spark microplate reader (Tecan).

## Phase contrast microscopy

The strains were grown to exponential phase and samples were collected. 1% final concentration of formaldehyde was used to fix the cells to be stored at 4°C. The cells were transferred to 1% agarose pads on glass slides and covered with cover slips for imaging. A T*i* eclipse inverted research microscope (Nikon) with 100×/1.45 numerical aperture (NA) objective (Nikon) was used to collect phase-contrast images. The images were processed and quantified using Fiji (ImageJ) and its MicrobeJ plugin.

## Generation and analysis of motility suppressors

Swimming motility plates containing the *lon-* strain after incubation at room temperature for 48 hours was exposed to UV irradiation for 0.4 to 1 second using the ChemiDoc imager under the setting: DNA gels – UV light. The plates were incubated for further one to two days and imaged again. The emerging suppressors were extracted from the outer edge of the suppressor zone using a pipette and plated on LB agar plates. They were grown overnight at 37°C, grown in overnight cultures and tested on motility agar plates to check if they are true suppressors. Six of the suppressors that showed motility equal to or more than the WT strain were selected and their genomic DNA was extracted using the Promega Wizard Genomic DNA Purification kit. The DNA from WT, *lon-* and Sup 1–6 was sent to SeqCenter, Pennsylvania, USA. The Illumina reads and variant calling revealed all the mutations present compared to the *P. aeruginosa* PAO1 whole genome NC_002516.2.

## Statistical analysis

All quantifications were made from arithmetic mean ± standard deviation of independent replicates. The value of replicate number, sample sizes and the statistical test used along with p-values are mentioned in the respective figure legends. The application Graphpad Prism Version 10.4.1 was used for the analyses and statistical tests.

## Supporting information

**S1 Fig. Lon levels and stability in the *lon* overexpression strain.** (A) Representative immunoblot of WT with empty vector (VC) and *lon* overexpression (WT $P_{ara}$ *lon*OE) using an anti-Lon antibody before and 90 min after induction using 0.2% arabinose. *Bottom* – Quantification of Lon relative to the VC from the immunoblots based on at least three independent replicates. (B) Growth curves of WT in the presence of 0, 500, 800 and 1000 μg/mL spectinomycin grown at 37°C and measured at $OD_{600}$. Means ± SD of three independent biological replicates plotted for each concentration. Spec – spectinomycin. (C) Representative immunoblot showing the change in Lon levels of the WT with empty vector (VC) and *lon* overexpression (WT $P_{ara}$ *lon*OE) over time after addition of 1 mg/ml spectinomycin to shut off protein synthesis and representative stain-free gel showing the total proteome of the proteomics samples.
(EPS)

**S2 Fig. *In vitro* degradation assays of β-casein, BioB and RlmN.** *In vitro* degradation assays testing Lon-dependent degradation of β-casein (A), BioB (B) and RlmN (C) in the presence and absence of ATP. Substrate concentrations used were 200 μg/mL for β-casein, 2.1 μM for BioB and RlmN. 0.4 μM of purified $Lon_6$ hexamer was added in each reaction. Quantifications of each of the degradation assays have been done using Image Lab 6.1 and plotted using the arithmetic mean ± SD of the quantified values of remaining substrate normalized relative to corresponding Lon levels for each time point. Each graph represents quantification of at least three independent experiments. One representative SDS-PAGE gel of the assay is shown for each protein. CK – creatine kinase.
(EPS)

**S3 Fig. Immunoblot showing Lon levels in wild type and *lon-* strains.** WT and *lon-* with empty vector (VC), *lon* overexpression (WT $P_{ara}$ *lon*) and *lon-* complemented with *lon* (*lon-* $P_{ara}$ *lon*) strains visualized using an anti-Lon antibody 0 and 90 min after induction with 0.2% arabinose.
(EPS)

**S4 Fig. Growth and pyocyanin levels of motility suppressor strains.** (A) Growth curves of WT, *lon-*, Sup3, Sup5 and Sup6 grown at 37°C and measured at $OD_{600}$. Means ± SD of three independent biological replicates plotted for each strain. (B) Pyocyanin produced by late exponential cultures grown at 37°C as visualized by the absorbance scan of supernatants of WT, *lon-*, Sup3, Sup5 and Sup6 strains from 500–900 nm. Means of three biological replicates plotted.
(EPS)

**S5 Fig. Quantification of microscopy and motility of strains expressing *sulA* and its mutants.** (A) Violin plot shows quantification of the phase-contrast microscopy images of the strains shown, grown with arabinose induction at 37°C to logarithmic phase. A total of 900 cells for each strain from a total of at least three biological replicates were used. Statistical significance was calculated using one-way ANOVA (Dunnett's multiple comparisons test) for pairs: WT VC vs WT $P_{ara}$ F-*sulA* and WT VC vs WT $P_{ara}$ F-*sulA*$^{D109V}$, $p < 0.0001$, ****; WT VC vs WT $P_{ara}$ F-*sulA*$^{H127L}$, $p = 0.7164$, (not significant, ns); WT + VC vs WT $P_{ara}$ F-*sulA*$^{fs}$, $p = 0.7439$, (not significant, ns). (B) Quantification of swimming motility in [Fig 5F]. Means of motility zone area ($cm^2$) ± SD of three independent biological replicates plotted. Statistical significance was calculated using one-way ANOVA (Dunnett's multiple comparisons test) for comparison of pairs: WT VC vs WT $P_{ara}$ F-*sulA* and WT VC vs WT $P_{ara}$ F-*sulA*$^{D109V}$, $p < 0.0001$, ****; WT VC vs WT $P_{ara}$ F-*sulA*$^{H127L}$, $p = 0.0006$, ***. (C) Swimming motility in soft agar without arabinose after 40–48 hours at 25°C for the strains shown. Means of motility zone area ($cm^2$) ± SD of three independent biological replicates are plotted. Statistical significance was calculated using one-way ANOVA (Dunnett's multiple comparisons test) for comparison of pairs: WT VC vs WT $P_{ara}$ F-*sulA*, $p = 0.0004$, ***; WT VC vs WT $P_{ara}$ F-*sulA*$^{H127L}$, $p = 0.7508$, (not significant, ns); WT VC vs WT $P_{ara}$ F-*sulA*$^{D109V}$, $p = 0.4674$, (not significant, ns). *Right* – Representative image of the swimming agar plate without arabinose.
(EPS)

**S6 Fig. Growth in piperacillin at different concentrations.** Growth curves of WT in the presence of 0, 0.25, 0.5 and 1 µg/mL piperacillin grown at 37°C and measured at $OD_{600}$. Means ± SD of three independent biological replicates are plotted for each condition.
(EPS)

**S1 Table. Strains used in this study.**
(PDF)

**S2 Table. Plasmids used in this study.**
(PDF)

**S3 Table. Primers used in this study.**
(PDF)

**S1 Data.** Quantitative proteomics (*lon*OE) source data, related to Fig 1.
(XLSX)

**S2 Data.** Quantitative proteomics (*lon*-) source data, related to Fig 3.
(XLSX)

## Acknowledgments

We thank members of the Jonas group for helpful discussions and feedback on the manuscript, the Hancock lab (University of British Columbia) for providing strains H103 and H1105 and the Clinical Proteomics Mass Spectrometry Core Facility at KI/KS for excellent service, support and advice.

## Author contributions

**Conceptualization:** Aswathy Kallazhi, Ute Römling, Kristina Jonas.

**Data curation:** Aswathy Kallazhi.

**Formal analysis:** Aswathy Kallazhi, Anamika Rahman.

**Funding acquisition:** Ute Römling, Kristina Jonas.

**Investigation:** Aswathy Kallazhi, Anamika Rahman.

**Methodology:** Aswathy Kallazhi, Ute Römling, Kristina Jonas.

**Project administration:** Kristina Jonas.

**Resources:** Ute Römling, Kristina Jonas.

**Supervision:** Aswathy Kallazhi, Ute Römling, Kristina Jonas.

**Validation:** Aswathy Kallazhi.

**Visualization:** Aswathy Kallazhi, Kristina Jonas.

**Writing – original draft:** Aswathy Kallazhi, Kristina Jonas.

**Writing – review & editing:** Aswathy Kallazhi, Ute Römling, Kristina Jonas.

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
