## [Decision Letter · Decision Letter 0]

Direct and indirect pathways linking the Lon protease to motility behaviors in the pathogen Pseudomonas aeruginosa

PLOS Pathogens

Dear Dr. Jonas,

Thank you for submitting your manuscript to PLOS Pathogens. After careful consideration, we feel that it has merit but does not fully meet PLOS Pathogens's publication criteria as it currently stands. Therefore, we invite you to submit a revised version of the manuscript that addresses the points raised during the review process.

Please submit your revised manuscript within 60 days May 06 2025 11:59PM. If you will need more time than this to complete your revisions, please reply to this message or contact the journal office at plospathogens@plos.org. Please include the following items when submitting your revised manuscript:

We look forward to receiving your revised manuscript.

Kind regards,

Vincent T Lee

Academic Editor

PLOS Pathogens

David Skurnik

Section Editor

Editor-in-Chief

PLOS Pathogens

orcid.org/0000-0003-2946-9497

Editor-in-Chief

PLOS Pathogens

orcid.org/0000-0002-7699-2064

**Additional Editor Comments:**

Thanks for submitting the manuscript. The reviewers identified these issues that should be addressed:

1. There is concern whether the use of spectinomycin leads to complete or partial inhibition. Determining this experimentally, such as a pulse chase, under the conditions used in this study would be good.

2. Statistics should be provided for Figure 3.

3. Determine the whether the protein level of the suppressor alleles of sulA

4. The inability to complement lonA from a plasmid for twitching and biofilm is curious, but no explanation is provided. Can different amount of inducer be tested to determine if the inability to complement is due to the amount of LonA made from the plasmid?

**Journal Requirements:**

At this stage, the following Authors/Authors require contributions: Aswathy Kallazhi, Anamika Rahman, and Ute Römling. Please ensure that the full contributions of each author are acknowledged in the "Add/Edit/Remove Authors" section of our submission form.

- ® on pages: 25 and 26.

- TM on pages: 25 and 26.

5) We have noticed that you have uploaded Supporting Information files, but you have not included a list of legends. Please add a full list of legends for your Supporting Information files after the references list.

Potential Copyright Issues:

- Please confirm that you are the photographer of Figure 4D, or provide written permission from the photographer to publish the photo under our CC BY 4.0 license.

**Reviewers' Comments:**

Reviewer's Responses to Questions

**Part I - Summary**

Reviewer #1: The ATP-dependent Lon protease plays vital roles in protein control across all life domains. Lon's absence causes multiple defects in motility, virulence, antibiotic tolerance, and biofilm formation in a pathogenic bacterium Pseudomonas aeruginosa. Through proteome analysis of Lon overexpression, authors identified new Lon substrate proteins, particularly those involved in motility and cell division. They confirmed several new substrates through in vitro protein degradation assay, including proteins crucial for flagella and pilus-based movement. Additionally, they discovered that Lon's degradation of SulA, a cell division inhibitor, is essential for proper cell division and motility. This work expands our understanding of how Lon protease regulation contributes to P. aeruginosa pathogenicity. While these findings are intriguing, the authors note that the manuscript requires further revisions in certain areas.

Reviewer #2: In their manuscript "Direct and indirect pathways linking the Lon protease to motility behaviors in the pathogen Pseudomonas aeruginosa", Kallazhi and colleagues describe the identification and characterisation of substrates of the Lon protease in Pseudomonas aeruginosa. They first perform quantitative proteomics to compare a strain overexpressing Lon to the matched WT strain from P. aeruginosa clone C, before and after inhibiting new protein synthesis with spectinomycin. This identifies 64 candidate substrates that satisfy 3 different analysis criteria. The authors then show that Lon can degrade nine of these proteins in vitro (two candidate substrates that were tested did not show in vitro degradation). Next, they investigate whether the steady state levels or stabilities of these substrates are affected by lon disruption in a different strain

(PAO1) and find that some of them are stabilised in the lon- strain, increasing the chances that they are bona fide substrates. Finally, they investigate phenotypes predicted to be affected by the new Lon substrates and find that motility, growth, biofilm production, and pigment production do in fact seem to be affected by Lon. They show that several of the phenotypes can be reverted by sulA suppressor mutations, but also that their lon- strain carries a background mutation in pilI, which likely impacts twitching motility and biofilm formation. Overall, these findings contribute to understanding of the roles of the Lon protease in P. aeruginosa regulation and physiology. The network of regulators in which Lon is embedded is complex, with many examples of crosstalk at multiple regulatory levels, and the observations presented here are interesting and useful. In particular, the observation that sulA suppressor mutations revert many lon- phenotypes is important. However, several complexities impact the interpretation of the data, and in some places, clarity could be improved.

Some suggestions for this are listed below:

1. There is good evidence at least in E. coli (PMID: 39003262), and to a lesser extent in P. aeruginosa (PMID: 32029587), that the AAA+ ATPase dependent protease families (Lon, ClpXP, HslUV, and FtsH) have some degree of redundancy or partially overlapping substrates. The abilities of these proteases to degrade overlapping subsets of substrate proteins seems like an important explanatory point for reconciling the results observed in the lon overexpression experiment with the results in the lon disruption experiment. This point is very briefly mentioned in the discussion, but it seems worth bringing it up earlier, and the assertion in the introduction that each protease "degrades a distinct panel of substrates" should be clarified to accommodate this evidence that their substrates and functions are overlapping.

2. The use of spectinomycin to inhibit new protein synthesis so that degradation rates could be assessed at the proteome level is useful for improving sensitivity to detect Lon impacts. However, spectinomycin has been described as an incomplete inhibitor of translation, that only slows but does not completely stop translocation. This does not necessarily invalidate it for use in this experiment - slowing the rate of new protein synthesis still can impart a relative increase on the importance of protein degradation in determining the levels of proteins. However, mentioning the possibility of some ongoing protein synthesis could potentially help explain some issues in the proteomics data: 1) substantial increases in abundance over time during the translation inhibition is observed for some proteins; and 2) the fold change from time 0 to time 30 is very similar to the fold change from 0 to time 60 for several of the candidate Lon substrates. If slow ongoing synthesis favoured some proteins over others, and/or if slow new synthesis could more or less keep pace with degradation after the initial decrease in new synthesis established a lower steady state level, these observations could be explained. Alternatively, artefacts of normalisation in the context of generally decreasing protein abundance during spectinomycin treatment could contribute to these issues. It would be good for the authors to comment on these patterns and the explanation(s) they favour.

3. The list of newly identified Lon substrates is interesting for many reasons. One reason is that several of them are known to play roles as chaperones or binding partners that contribute to protein stabilisation, folding, and degradation, raising the possibility that there could be complex feedback effects. IbpA is one example - it is a chaperone of unfolded proteins that is upregulated by heat shock and starvation. Could disruption of proteostasis through Lon dysregulation affect the availability of unfolded proteins for IbpA to bind, and thus affect its own availability for degradation? The flagellar synthesis cascade is another example of this: FliS is a chaperone for FliC secretion; FlgE is required for FlgM secretion, and FlgM is the chaperone/anti-sigma factor for FliA, which is also the sigma factor driving FliC expression. Several of these proteins have intrinsically disordered regions on their own. Does Lon do a job of degrading any of these proteins that are in excess stoichiometrically and not bound to their proper binding partners? Such a homeostasis-promoting role could explain the lack of disruption to steady state levels in the lon- strain and the ability of the sulA suppressor mutation to completely alleviate the lon- strain's motility defect.

4. The discovery of a background mutation in the lon- strain which might explain the biofilm and twitching phenotypes seems like an important point. Is there selection in lon- strains for type IV pilus mutations? Do you think that previously reported biofilm and twitching phenotypes are also likely to be due to background mutations? Mutations to type IV pilus genes are very common in clinical isolates and seem to confer advantages in a variety of infection-mimicking conditions; perhaps they can also alleviate stress imposed by lon mutation.

5. Since it is mentioned in the introduction as a previously identified substrate of Lon in P. aeruginosa, it might be worth commenting on the status of Hfq in these proteomics data sets.

**Part II – Major Issues: Key Experiments Required for Acceptance**

Reviewer #1: Major concerns

1. Although this manuscript identifies SulA degradation by Lon protease as a key finding, similar mechanisms have already been characterized in other bacterial species. Therefore, the manuscript needs to clearly highlight the novel aspects specific to P. aeruginosa and demonstrate how they differ from previously established mechanisms in other bacteria.

2. Line 217-218: to verify the mechanism of compensated phenotypes, authors should either analyze transcriptional regulation using specific transcriptional regulators, or alternatively, treat samples with rifampicin (a transcriptional inhibitor) to exclude the possibility of transcriptional regulation.

3. Figure 3: The authors suggested increased stability of FliG, FliS, FlgE, and RpoN in the lon mutant strain compared to wild-type. However, this comparison lacks statistical analysis to validate these differences. Furthermore, statistical analyses are missing from multiple figures throughout the manuscript, which is necessary to support the significance of the reported findings.

4. Line 250-252: The twitching and biofilm phenotypes fail to recover in the lon mutant strain, unlike other phenotypes. The authors should address this discrepancy either by testing alternative plasmid systems, or if this approach is not feasible, provide a detailed explanation for why these two phenotypes differ from the others in their recovery pattern.

5. While the authors mentioned that the mechanistic basis of Lon's effects on these processes remains poorly understood in Discussion part, this manuscript does not sufficiently address the underlying molecular mechanisms of Lon protease activity. To provide deeper mechanistic insights, the authors should conduct additional experiments including analysis of substrate degron sequences, interaction assay with Lon-accessory proteins, and substrate degradation assays using various protease mutant strains

6. The manuscript does not explain the mechanism by which SulA regulates motility in P. aeruginosa. If experimental evidence is not available, the authors should propose a testable hypothesis to explain this regulatory relationship

Reviewer #2: Ideally, the role of the pilI background mutation in the biofilm and twitching motility defects of the lon- should be directly tested by recreating the lon disruption in a clean PAO1 background. Alternatively, perhaps an existing pilI mutant could be tested for these phenotypes to see if the lon- strain is phenocopied with respect to biofilm formation and twitching motility? It would be interesting to know whether other lon- strains with reported biofilm defects also have type IV pilus mutations in their backgrounds, but this is outside the scope of this manuscript. However, highlighting this observation here may motivate other researchers to check the backgrounds of their lon- strains.

**Part III – Minor Issues: Editorial and Data Presentation Modifications**

Reviewer #1: Minor concerns

1. The distinction between 'significantly degraded' and 'lower stability' in lonOE strain shown in Figure 1C requires clarification, as protein degradation rate directly correlates with protein stability. This terminology should be explicitly defined and distinguished in both the main text and figure legend to avoid confusion.

Reviewer #2: 1. The point mutation in lon- strain should probably be listed as part of the genotype(s) in the strain table for clarity.

2. The y-axes of some panels in Fig. 1B appear mislabeled.

3. The author's statement suggests that proteases are promising drug targets (line 34). Some would argue that the AAA+ ATPase dependent proteases are very challenging drug targets. Highly conserved homologs exist in the human mitochondria for several of these proteases, making it very difficult to selectively target the bacterial proteins. Also, where small molecule inhibitors or activators of these proteins have been found, they have had poor pharmacological properties and substantial efforts to improve these properties have failed.

4. Ideally, raw proteomics data should be uploaded to a public repository such as PRIDE.

PLOS authors have the option to publish the peer review history of their article (what does this mean? ). If published, this will include your full peer review and any attached files.

**Do you want your identity to be public for this peer review?** For information about this choice, including consent withdrawal, please see our Privacy Policy .

Reviewer #1: No

Reviewer #2: No

**Figure resubmission:**

**Reproducibility:**



---

## [Decision Letter · Decision Letter 1]

Dear Dr Jonas,

We are pleased to inform you that your manuscript 'Direct and indirect pathways linking the Lon protease to motility behaviors in the pathogen Pseudomonas aeruginosa' has been provisionally accepted for publication in PLOS Pathogens.

Best regards,

Vincent T Lee

Academic Editor

PLOS Pathogens

David Skurnik

Section Editor

PLOS Pathogens

Sumita Bhaduri-McIntosh

Editor-in-Chief

PLOS Pathogens

orcid.org/0000-0003-2946-9497

Michael Malim

Editor-in-Chief

PLOS Pathogens

orcid.org/0000-0002-7699-2064

Congratulations!

Thank you very much for thoughtfully addressing the reviewers' concerns and providing insightful responses. I hope you also find that this process improved your manuscript.

Reviewer Comments (if any, and for reference):

Reviewer's Responses to Questions

**Part I - Summary**

Reviewer #1: Authors respond properly according to reviewer and editor's comments.

Reviewer #2: This revised manuscript describes, as did the original, the identification of substrates of the Lon protease in Pseudomonas aeruginosa. It shows that most of the previously described phenotypes of lon mutations can be ascribed to stabilisation of one substrate, the cell division inhibitor SulA. In response to the first round of reviews, the authors have added a new experiment showing that the lon- phenotypes which could not be complemented by lon overexpression were instead complemented by overexpression of pilI, which compensated for a pilI background mutation in the lon- strain. They have also improved the clarity of the manuscript in several places and added statistical analyses in some figures where they were previously lacking. These changes strengthen the manuscript. Overall, this manuscript contributes useful knowledge for microbiologists studying proteostasis and anyone interested in Pseudomonas aeruginosa physiology.

**Part II – Major Issues: Key Experiments Required for Acceptance**

Reviewer #1: Authors respond properly according to reviewer and editor's comments.

Reviewer #2: My concerns have been addressed.

**Part III – Minor Issues: Editorial and Data Presentation Modifications**

Reviewer #1: Authors respond properly according to reviewer and editor's comments.

Reviewer #2: My concerns have largely been addressed. The authors have argued that their assertion that intracellular AAA ATPase proteases are promising drug targets is accurate. I think that this assessment should be qualified by multiple caveats, but the manuscript has nothing to do with drug discovery, so our somewhat differing opinions on this matter are not important.

PLOS authors have the option to publish the peer review history of their article (what does this mean? ). If published, this will include your full peer review and any attached files.

**Do you want your identity to be public for this peer review?** For information about this choice, including consent withdrawal, please see our Privacy Policy .

Reviewer #1: No

Reviewer #2: No

---

## [Editor Report · Acceptance letter]

Dear Dr Jonas,

We are delighted to inform you that your manuscript, "Direct and indirect pathways linking the Lon protease to motility behaviors in the pathogen Pseudomonas aeruginosa," has been formally accepted for publication in PLOS Pathogens.

Best regards,

Sumita Bhaduri-McIntosh

Editor-in-Chief

PLOS Pathogens

orcid.org/0000-0003-2946-9497

Michael Malim

Editor-in-Chief

PLOS Pathogens

orcid.org/0000-0002-7699-2064